# Sculpting User Preferences for Recommendation with Positive-Negative Diffusion Guidance

## Abstract

Diffusion models are emerging as a powerful generative paradigm for sequential recommendation, demonstrating a remarkable ability to model complex user-item interaction dynamics. Despite their strong modeling ability, most diffusion-based recommenders face limited generative control because the standard classifier-free guidance derives its repulsive signal from a global and user-agnostic unconditional prior, which prevents the model from directly exploiting negative feedback at inference. A natural solution is to replace the unconditional prior with user-aware negative conditions. However, this is challenging because, unlike in text-to-image tasks where negative prompts acquire stable semantics from a pre-trained text encoder, item embeddings in recommendation are learned dynamically. As a result, a "negative condition" is not guaranteed to provide effective repulsive guidance unless the model is explicitly trained to recognize it as a signal for avoidance. To enable effective and steerable negative guidance in diffusion recommenders, we propose SteerRec, a novel framework built upon two core innovations. At inference, we introduce Positive-Negative Guidance (PNG) inference mechanism, which replaces the generic unconditional prior with a user-aware negative condition. To ensure the negative condition provides meaningful repulsive guidance in the dynamic embedding space, we design a Guidance Alignment Triplet Loss (GAL). The GAL is a margin-based objective that explicitly aligns the training process with PNG by ensuring the model's prediction under a positive condition is closer to the target item than its prediction under a negative condition. Extensive experiments on three widely used public benchmarks provide strong empirical evidence for the effectiveness of SteerRec. Our implementation is available at https://anonymous.4open.science/r/SteerRec-5D70.

## 1 Introduction

Modern recommender systems learn user preferences from historical interactions to rank relevant items from large-scale catalogs, and within this landscape sequential recommendation has become a critical subfield for modeling temporal user dynamics to predict the next item. (Hidasi et al., 2016; Kang & McAuley, 2018; Sun et al., 2019; Zhou et al., 2020; Xie et al., 2022). Since user preferences evolve with context and often follow complex distributions, generative modeling offers a natural and powerful paradigm for sequential recommendation. Diffusion models (DMs) (Sohl-Dickstein et al., 2015; Ho et al., 2020), owing to their ability to capture intricate preference distributions and iteratively refine predictions, have emerged as a backbone for generative recommendation (Rajput et al., 2023; Wu et al., 2024; Wang et al., 2023a; Zhao et al., 2024; Mao et al., 2025).

Existing diffusion-based sequential recommenders typically formulate the task as conditional generation, synthesizing an embedding vector for the user's next preferred item. Most conditional diffusion models adopt classifier-free guidance (CFG) as the standard inference mechanism, in which the conditional prediction is extrapolated away from the unconditional prediction. This guidance stabilizes the diffusion process and improves the fidelity of the generated outputs. Early approaches such as DreamRec conditioned generation on chronological user history, relying exclusively on positive signals (Yang et al., 2023; Li et al., 2023; Wang et al., 2024). Recognizing the critical role of negative signals in shaping user preferences (Chen et al., 2023; Zhang et al., 2024), PreferDiff ad-

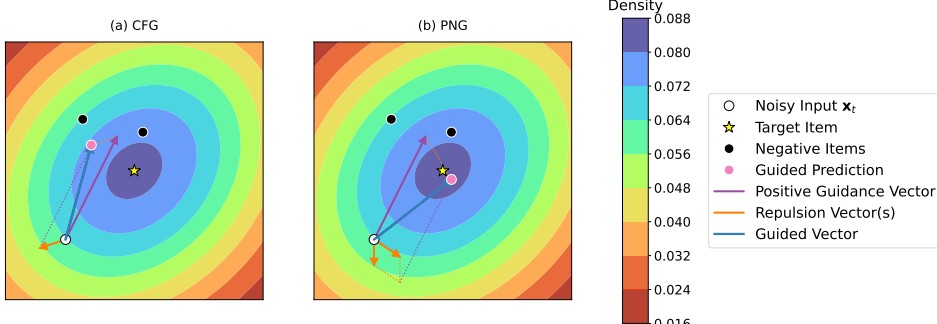

Figure 1: Illustration of the guidance mechanism within a single denoising step at time $t$. Heatmaps show the user's preference density. (a) CFG: Contrasting with a global prior creates a single, non-personalized repulsive force. (b) SteerRec (PNG): Contrasting with user-aware negatives creates targeted repulsive forces.

vanced the field by incorporating negative samples through Bayesian Personalized Ranking (BPR) to strengthen the learning signal (Rendle et al., 2009; Liu et al., 2025). Although incorporating negative samples via a ranking objective enriches the embedding space, the role of these signals is confined to the training loss. Consequently, these signals are not used as an explicit steering force during the inference-time denoising process. Using negative signals only in the training loss is an indirect way of handling negative feedback, failing to unlock the native guidance capabilities and full generative potential of DMs.

This indirect handling of negative signals is the root cause of a significant training-inference discrepancy in prior diffusion recommenders. The issue stems from their continued reliance on the standard CFG mechanism. In CFG, the repulsive force originates from the model's prediction conditioned on the null context ($\emptyset$), an output that serves as a user-agnostic prior for every user. This "one-size-fits-all" approach, illustrated in Figure 1(a), is ineffective for targeted avoidance, which means the guided prediction can still land undesirably close to disliked items. Consequently, a fundamental misalignment arises: negative signals are used to shape the embedding space during training but cannot be used to directly steer the generative process away from undesirable items at inference time.

Thus, an important research question emerges: *How can we effectively incorporate user-aware negative information as a direct guidance signal during inference?* Addressing this question requires moving beyond the standard CFG framework, which is not designed for negative conditioning (Ho & Salimans, 2022). Interestingly, in the field of text-to-image generation, negative guidance has proven to be highly effective (Rombach et al., 2022; Gandikota et al., 2023; Bansal et al., 2023; Ban et al., 2024; Koulischer et al., 2025). Its success, however, is largely attributable to the availability of large-scale and pre-trained semantic encoders, rather than explicit training objectives that enforce repulsion. For example, to generate "a man without a beard", a user can supply a positive prompt ("a man") alongside a negative prompt ("beard"). Since powerful encoders such as CLIP provide stable and universal representations for both concepts, the denoising network is thus able to distinguish what to generate and what to avoid. By replacing the unconditional (null prompt) prediction with the prediction conditioned on the negative prompt ("beard"), CFG is cleverly adapted to repulse the undesired concept ("beard") while steering the generation toward the desired one ("a man"). However, this negative guidance paradigm does not transfer directly to recommendation. Unlike text-to-image tasks where conditions are expressed in natural language and grounded in a fixed semantic space, recommendation operates in a learned and evolving embedding space where positive conditions (user history), negative conditions, and target items are all drawn from the same item set (Koren et al., 2009). As a result, their semantics are relative and interdependent, rather than fixed and universally interpretable. This makes negative guidance in recommendation inherently unstable: simply providing a negative item at inference does not guarantee meaningful repulsion, because the model has not been trained to interpret this negative signal as a repulsive force. In other words, the absence of an externally grounded semantic space (like the role of CLIP in vision-language tasks)

means that recommendation systems require an explicit training objective to align inference-time negative conditions with effective guidance.

To address this challenge, we introduce a new framework named SteerRec, which integrates negative signals directly into inference-time guidance and aligns the training process accordingly. As illustrated in Figure 1(b), SteerRec enforces both attraction toward desired items and targeted repulsion from negatives by contrasting the positive condition against a user-aware negative condition. Specifically, SteerRec introduces a Positive-Negative Guidance (PNG) mechanism, which replaces the user-agnostic unconditional prior with instance-specific negative conditions and admits a principled likelihood-ratio interpretation (Neyman & Pearson, 1933; Casella & Berger, 2024). To ensure the effectiveness of the PNG mechanism in the dynamic embedding space, we further design a Guidance Alignment Triplet Loss (GAL). GAL is a margin-based objective designed to structure the denoising network's output space (Schroff et al., 2015; Sohn, 2016; He et al., 2020; Sun et al., 2020). GAL enforces a triplet-based geometric constraint: for a ground-truth item (the anchor), the model's prediction under a positive condition must be closer to the anchor than its prediction under a negative condition. This explicit alignment during training empowers the PNG mechanism to exert precise and reliable repulsive control during inference. Our contributions are summarized as follows:

- We propose SteerRec, a novel diffusion recommendation framework that enables direct and reliable negative guidance. SteerRec resolves a critical training-inference misalignment in existing methods by replacing the user-agnostic unconditional prior of CFG with user-aware negative conditions.

- By introducing a Positive-Negative Guidance mechanism at inference and a complementary Guidance Alignment Triplet Loss during training, SteerRec effectively structures the dynamic embedding space. This process leads to more precise and controllable user preference generation.

- Extensive experiments on three public benchmark datasets demonstrate the effectiveness and superiority of SteerRec, with significant performance gains over leading baselines. Our in-depth analyses further validate the advantages of our direct negative guidance paradigm.

## 2 PRELIMINARY

This section provides the technical background necessary to understand our proposed method. We formally define the sequential recommendation task and detail the core mechanics of DMs and CFG, the latter of which motivates our work.

### 2.1 SEQUENTIAL RECOMMENDATION

The task of sequential recommendation is to predict the next item a user will interact with based on their interaction history. Formally, for a set of users $\mathcal{U}$ and items $\mathcal{I}$, the goal is to predict the next item $i_n \in \mathcal{I}$ for a user $u \in \mathcal{U}$ with an interaction history of $S_u = (i_1, i_2, \ldots, i_{n-1})$. In mainstream discriminative frameworks, this is operationalized as a ranking task. Each item $i$ is represented by a learnable embedding vector $\mathbf{x} \in \mathbb{R}^d$. The sequence of embeddings $(\mathbf{x}_1, \mathbf{x}_2, \ldots, \mathbf{x}_{n-1})$ is processed by a neural encoder (e.g., a Transformer) to produce a single context vector $\mathbf{c}^+ \in \mathbb{R}^d$. The model is trained with a ranking objective to distinguish the true next item $i_n$ from a set of sampled negative items $H \subset \mathcal{I}$. The goal is to learn representations that ensure the score computed from $\mathbf{c}^+$ for item $i_n$ is higher than for any item $j \in H$. The final recommendation list is generated by ranking all candidate items based on their computed scores.

### 2.2 DIFFUSION MODELS FOR SEQUENTIAL RECOMMENDATION

DMs have recently been applied to sequential recommendation, marking a shift from discriminative paradigms to a generative approach. Instead of learning to classify the next item from a set of candidates, the task is reframed as generating an embedding that represents the user's next preferred item, conditioned on their history (Yang et al., 2023; Liu et al., 2025; Li et al., 2023). This process is typically defined by two complementary stages:

**Forward Process.** The forward process is a fixed Markov chain that gradually injects Gaussian noise into the ground-truth target item embedding $\mathbf{x}_0$ over $T$ timesteps. This is governed by a predefined variance schedule $\{\beta_t\}_{t=1}^T$.

$$q(\mathbf{x}_{1:T}|\mathbf{x}_0) = \prod_{t=1}^{T} q(\mathbf{x}_t|\mathbf{x}_{t-1}), \quad \text{where} \quad q(\mathbf{x}_t|\mathbf{x}_{t-1}) = \mathcal{N}(\mathbf{x}_t; \sqrt{1-\beta_t}\mathbf{x}_{t-1}, \beta_t\mathbf{I}) \tag{1}$$

A key property of this process is that the noisy latent $\mathbf{x}_t$ can be sampled in a closed form for any timestep $t$:

$$\mathbf{x}_t = \sqrt{\bar{\alpha}_t}\mathbf{x}_0 + \sqrt{1-\bar{\alpha}_t}\epsilon, \quad \text{where} \quad \epsilon \sim \mathcal{N}(\mathbf{0}, \mathbf{I}) \tag{2}$$

where $\alpha_t = 1 - \beta_t$ and $\bar{\alpha}_t = \prod_{s=1}^{t} \alpha_s$. As $t \to T$, $\mathbf{x}_T$ converges to an isotropic Gaussian.

**Reverse Process.** The reverse process aims to learn the data distribution by approximating the true posterior $q(\mathbf{x}_{t-1}|\mathbf{x}_t, \mathbf{x}_0)$ with a parameterized model $p_\theta(\mathbf{x}_{t-1}|\mathbf{x}_t, \mathbf{c}^+)$. This is achieved by training a neural network $F_\theta$ to denoise the corrupted input $\mathbf{x}_t$, conditioned on the user's historical context vector $\mathbf{c}^+$. While the full training objective involves optimizing a variational bound on the log-likelihood, it can be simplified to a mean squared error objective (Ho et al., 2020). Following recent diffusion recommenders (Yang et al., 2023; Liu et al., 2025), we parameterize our model to predict the original data $\mathbf{x}_0$ rather than the noise term $\epsilon$. This offers the conceptual advantage of directly aligning the network's output with the task's ultimate goal. The model $F_\theta(\mathbf{x}_t, \mathbf{c}^+, t)$ is thus optimized via the following simple reconstruction loss, $L_{\text{recon}}$:

$$L_{\text{recon}}(\theta) = \mathbb{E}_{\mathbf{x}_0, \mathbf{c}^+, t}\left[\|\mathbf{x}_0 - F_\theta(\mathbf{x}_t, \mathbf{c}^+, t)\|^2\right] \tag{3}$$

**Inference and Recommendation.** At inference time, the model generates the next-item embedding through an iterative denoising process. Starting from pure Gaussian noise $\mathbf{x}_T \sim \mathcal{N}(\mathbf{0}, \mathbf{I})$, the model iteratively applies the reverse process for $t = T, \ldots, 1$. In each step, the denoising network $F_\theta(\mathbf{x}_t, \mathbf{c}^+, t)$ predicts the clean embedding, and an efficient sampler such as Denoising Diffusion Implicit Models (DDIM) (Song et al., 2021a) is used to estimate the next state $\mathbf{x}_{t-1}$. After $T$ steps, this process yields the final generated embedding $\hat{\mathbf{x}}_0$. Finally, this embedding is used to rank all candidate items by computing their inner product scores with $\hat{\mathbf{x}}_0$, and the top-K highest-scoring items are returned as the recommendation list.

### 2.3 CONTROLLABLE GENERATION WITH CFG

To enhance the influence of the conditioning signal $\mathbf{c}^+$ on the generative process, diffusion-based recommenders commonly adopt CFG (Ho & Salimans, 2022). The core idea is to train a single network $F_\theta$ to operate in both a conditional mode (receiving $\mathbf{c}^+$) and an unconditional mode. This is achieved via conditional dropout, where during training, the context $\mathbf{c}^+$ is randomly replaced by a shared, learnable null context token $\emptyset$.

This dual-mode training enables control at inference time. At inference, the guided prediction is formed by combining the model's outputs under both its conditional and unconditional modes. By extrapolating away from the unconditional prediction, the conditional signal can be amplified. The guided prediction of the clean data, $\hat{\mathbf{x}}_0$, is thus formulated as:

$$\hat{\mathbf{x}}_0(\mathbf{x}_t, \mathbf{c}^+) = (1+w) \cdot F_\theta(\mathbf{x}_t, \mathbf{c}^+, t) - w \cdot F_\theta(\mathbf{x}_t, \emptyset, t) \tag{4}$$

where $w$ is the guidance scale. A higher $w$ value strengthens the effect of the condition, which is known to improve sample fidelity at the potential cost of diversity (Dhariwal & Nichol, 2021). This guidance mechanism steers the generation away from the generic, marginal distribution represented by the unconditional prediction, which lacks the specificity required for fine-grained, personalized negative feedback.

## 3 METHODOLOGY

Our SteerRec overcomes the limitations of standard diffusion recommenders through a fundamental redesign of both the inference-time guidance and the training objective. Its core lies in a new inference paradigm, Positive-Negative Guidance, which replaces the generic unconditional prior with

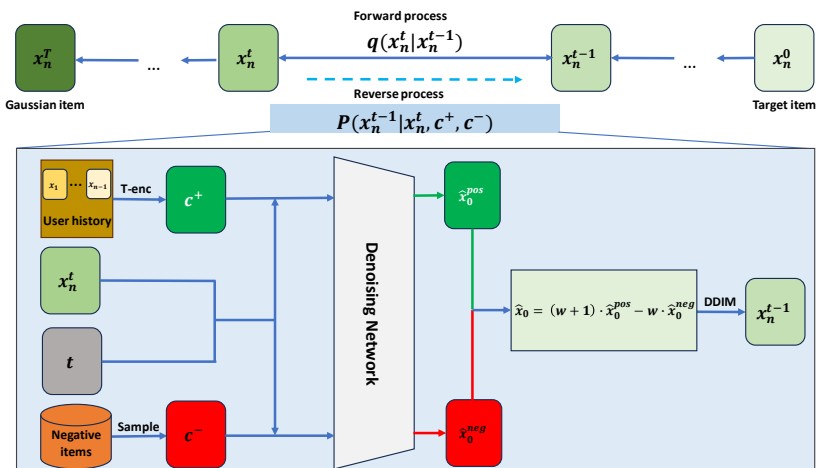

Figure 2: The architecture of the SteerRec framework. The top panel illustrates the overall forward and reverse diffusion processes. The bottom panel details a single diffusion step under our proposed PNG mechanism. Within this step, a positive condition ($\mathbf{c}^+$) is encoded from user history, while a negative condition ($\mathbf{c}^-$) is formed from items sampled from the user's uninteracted items. The denoising network separately utilizes the positive ($\mathbf{c}^+$) and negative ($\mathbf{c}^-$) conditions to generate two predictions, $\hat{\mathbf{x}}_0^{\text{pos}}$ and $\hat{\mathbf{x}}_0^{\text{neg}}$. These predictions are then combined by the PNG guidance formula to produce a guided prediction $\hat{\mathbf{x}}_0$ for the DDIM update step. This PNG mechanism is enabled by our GAL used during training.

user-aware negative feedback. To ensure this new guidance mechanism is effective, we introduce a corresponding Guidance Alignment Triplet Loss during training, which explicitly aligns the denoising network's behavior with the PNG mechanism. In the following sections, we will detail each of these components and discuss the strategies for constructing the negative conditions that power this framework. The overall architecture of SteerRec is illustrated in Figure 2.

### 3.1 THE POSITIVE-NEGATIVE GUIDANCE PARADIGM

The core of our methodology is an inference paradigm that replaces the "positive-vs-unconditional" structure of CFG with a more precise "positive-vs-negative" guidance mechanism. This allows the generative process to be directly steered by user-aware negative feedback.

**PNG Formulation.** Our primary goal is to directly leverage user-aware negative feedback as a guiding signal within the denoising process. The standard CFG provides a valuable extrapolation principle, but the repulsive force of this framework is user-agnostic, steering generation away from a generic prior instead of specific items a user dislikes. Inspired by CFG's extrapolation, we introduce the PNG mechanism, which replaces the generic prior with a user-aware negative condition. The PNG framework is adapted for our denoising network $F_\theta$ that predicts the original data $\mathbf{x}_0$, and is formalized as:

$$\hat{\mathbf{x}}_0(\mathbf{x}_t, \mathbf{c}^+, \mathbf{c}^-) = (1+w) \cdot F_\theta(\mathbf{x}_t, \mathbf{c}^+, t) - w \cdot F_\theta(\mathbf{x}_t, \mathbf{c}^-, t) \quad (5)$$

where $\mathbf{c}^+$ is the positive condition derived from user history, $\mathbf{c}^-$ (detailed in Section 3.3) is the user-aware negative condition, and $w$ is the guidance scale that controls the guidance intensity, balancing fidelity to the positive condition against recommendation diversity. For an intuitive understanding, Equation 5 can be rewritten as:

$$\hat{\mathbf{x}}_0(\mathbf{x}_t, \mathbf{c}^+, \mathbf{c}^-) = F_\theta(\mathbf{x}_t, \mathbf{c}^+, t) + w \cdot \left( F_\theta(\mathbf{x}_t, \mathbf{c}^+, t) - F_\theta(\mathbf{x}_t, \mathbf{c}^-, t) \right) \quad (6)$$

Here, the difference between the positive and negative predictions serves as a corrective guidance vector, actively pushing the prediction away from the space defined by the negative condition.

**Theoretical Foundation.** Our guidance rule is not merely heuristic but is theoretically grounded in the principles of score-based modeling (Song & Ermon, 2019; 2020; Song et al., 2021b). The guided

reverse process, as implemented through Eq. 5, possesses an instantaneous score $(\nabla_{\mathbf{x}_t} \log p^*(\mathbf{x}_t))$ that is equivalent to the gradient of a likelihood-ratio-tilted density:

$$p^*(\mathbf{x}_t | \mathbf{c}^+, \mathbf{c}^-) \propto \frac{p_\theta(\mathbf{x}_t | \mathbf{c}^+)^{1+w}}{p_\theta(\mathbf{x}_t | \mathbf{c}^-)^w} \tag{7}$$

This target density is motivated by the Neyman-Pearson Lemma (Neyman & Pearson, 1933), which identifies the likelihood ratio as the optimal statistic for discriminating between the positive condition $\mathbf{c}^+$ and the negative condition $\mathbf{c}^-$. Therefore, our guidance steers the reverse process towards this target distribution at each noise level $t$. A detailed derivation showing the equivalence in score-space and its connection to our $\mathbf{x}_0$-prediction model is provided in Appendix C.

**Efficient Reverse Process with DDIM.** To efficiently generate the final item embedding, we integrate our guidance rule into the DDIM sampling process. Unlike the original DDPM sampler, DDIM enables a much faster reverse process by defining a non-Markovian chain that permits a small number of large, deterministic sampling steps. The one-step update from $\mathbf{x}_t$ to $\mathbf{x}_{t-1}$ proceeds deterministically by first computing our guided prediction $\hat{\mathbf{x}}_0$ (from Eq. 5) and then using it to solve for the next state:

$$\mathbf{x}_{t-1} = \sqrt{\bar{\alpha}_{t-1}} \hat{\mathbf{x}}_0 + \sqrt{1 - \bar{\alpha}_{t-1}} \cdot \left( \frac{\mathbf{x}_t - \sqrt{\bar{\alpha}_t} \hat{\mathbf{x}}_0}{\sqrt{1 - \bar{\alpha}_t}} \right) \tag{8}$$

This process is iterated for a small number of steps to generate the final embedding.

## 3.2 TRAINING-INFERENCE ALIGNMENT

The effectiveness of our PNG inference mechanism hinges on the denoising network $F_\theta$ producing semantically distinct outputs for the positive ($\mathbf{c}^+$) and negative ($\mathbf{c}^-$) conditions. If trained only with a simple reconstruction objective, the network has no incentive to interpret the negative condition $\mathbf{c}^-$ as a repulsive signal, which undermines the effectiveness of the PNG mechanism. To resolve this training-inference discrepancy, we must explicitly teach the model the oppositional nature of these conditions.

To achieve this, we introduce the Guidance Alignment Triplet Loss, a margin-based objective inspired by deep metric learning (Schroff et al., 2015; Sohn, 2016). The core principle of GAL is to structure the model's output space such that the prediction under the positive condition is geometrically closer to the ground-truth item than the prediction under the negative condition is. To formalize this, given a noisy input $\mathbf{x}_t$, we first compute two denoised predictions under the opposing conditions:

$$\hat{\mathbf{x}}_0^{\text{pos}} = F_\theta(\mathbf{x}_t, \mathbf{c}^+, t) \quad \text{(the positive prediction)} \tag{9}$$

$$\hat{\mathbf{x}}_0^{\text{neg}} = F_\theta(\mathbf{x}_t, \mathbf{c}^-, t) \quad \text{(the negative prediction)} \tag{10}$$

Based on these two predictions, GAL is formulated as:

$$L_{\text{GAL}} = \max(0, d(\hat{\mathbf{x}}_0^{\text{pos}}, \mathbf{x}_0^+) - d(\hat{\mathbf{x}}_0^{\text{neg}}, \mathbf{x}_0^+) + m) \tag{11}$$

where $d(\cdot, \cdot)$ is a distance metric (e.g., Cosine distance), and $m$ is a positive margin hyperparameter that defines the minimum desired separation between the distances.

While GAL enforces the necessary alignment for guidance, we still require an objective to ensure the generated item is accurate. This is accomplished by a standard reconstruction loss, which encourages the positive prediction to be close to the ground truth:

$$L_{\text{recon}} = d(\hat{\mathbf{x}}_0^{\text{pos}}, \mathbf{x}_0^+) \tag{12}$$

The final training objective, $L$, is a composite loss that combines both alignment and reconstruction:

$$L = (1 - \mu) \cdot L_{\text{recon}} + \mu \cdot L_{\text{GAL}} \tag{13}$$

Here, $\mu \in [0, 1]$ is a hyperparameter that balances the contribution of the reconstruction objective (generative fidelity) and the alignment objective (guidance effectiveness).

## 3.3 INSTANTIATING NEGATIVE CONDITIONS

A crucial component of SteerRec is the construction of the negative condition $\mathbf{c}^-$. Unlike the static, global prior in CFG, our negative condition is dynamic and instance-specific. We instantiate this condition by sampling a set of negative items and aggregating their corresponding embeddings.

**Negative Item Sampling.** We adopt two distinct but complementary sampling strategies to maintain computational efficiency and effectiveness during the training and inference phases. During training, we employ in-batch negative sampling, treating all other items in a mini-batch as negative samples for each positive instance. This strategy, widely adopted in self-supervised learning (Chen et al., 2020; Karpukhin et al., 2020), is highly efficient and provides a diverse set of challenging negatives for each training instance. At inference time, when batch information is unavailable, we construct the negative condition by sampling a set of items randomly from the global item corpus. Even this simple random sampling strategy provides a robust repulsive signal, effectively validating the potential of negative guidance in diffusion-based recommendation.

**Embedding Aggregation.** To generate a stable and comprehensive repulsive signal, we aggregate the set of selected $N_{neg}$ negative item embeddings $\{\mathbf{x}_{neg}^{(k)}\}_{k=1}^{N_{neg}}$ into a single condition vector $\mathbf{c}^-$ using the centroid method (Xie et al., 2016), as follows:

$$\mathbf{c}^- = \frac{1}{N_{neg}} \sum_{k=1}^{N_{neg}} \mathbf{x}_{neg}^{(k)} \tag{14}$$

While global random sampling serves as an efficient baseline strategy, the performance of SteerRec can be further unlocked when higher-quality sources of negative feedback are available, as demonstrated in Appendix F. The complete training and inference procedures are detailed in Appendix D.2.

# 4 EXPERIMENTS

In this section, we conduct extensive experiments to answer the following research questions:

- **RQ1:** How does SteerRec perform compared with other sequential recommenders?
- **RQ2:** How do the core components of SteerRec, the PNG inference mechanism and the GAL objective, each contribute to its overall performance?
- **RQ3:** How do SteerRec's key hyperparameters influence its performance, and what is its training efficiency?

## 4.1 EXPERIMENTAL SETUP

**Datasets and Baselines.** We evaluate SteerRec on three public Amazon Review datasets: Sports and Outdoors, Beauty, and Toys and Games. Following the protocol of recent works (Liu et al., 2025), we adopt a chronological 80/10/10 user-based split and perform five-core filtering. We compare SteerRec against a comprehensive suite of baselines, including traditional sequential models (e.g., GRU4Rec (Hidasi et al., 2016), SASRec (Kang & McAuley, 2018), BERT4Rec (Sun et al., 2019)), advanced paradigms such as contrastive learning (e.g., CL4SRec (Xie et al., 2022)) and autoregressive generation (e.g., TIGER (Rajput et al., 2023)), and other diffusion-based methods (e.g., DiffuRec (Li et al., 2023), DreamRec (Yang et al., 2023), PreferDiff (Liu et al., 2025)). Detailed dataset statistics and baseline descriptions are provided in Appendix D.

**Implementation and Evaluation.** To ensure a fair comparison, we implement SteerRec using the same SASRec backbone and key hyperparameters (e.g., embedding dimension of 3072) as recent diffusion recommenders (Liu et al., 2025). We use a linear noise schedule with the deterministic DDIM sampler for efficient inference. For evaluation, we adopt two standard top-K ranking metrics, Recall@K and NDCG@K (K={5, 10}), computed in a full-ranking setting over the entire item corpus. Further implementation details, including the hyperparameter search space, are available in Appendix D.2.

## 4.2 OVERALL PERFORMANCE COMPARISON (RQ1)

As shown in Table 1, diffusion-based models generally outperform traditional sequential models, which can be attributed to their powerful ability to model complex data distributions. Among all baselines, our proposed SteerRec achieves the best performance across all datasets and metrics. The

Table 1: Overall performance comparison on all three datasets. The best performance is in **bold**, and the second best is underlined. All improvements of SteerRec are statistically significant ($p \ll 0.05$).

| Model | Sports and Outdoors | | | | Beauty | | | | Toys and Games | | | |
|---|---|---|---|---|---|---|---|---|---|---|---|---|
| | R@5 | N@5 | R@10 | N@10 | R@5 | N@5 | R@10 | N@10 | R@5 | N@5 | R@10 | N@10 |
| GRU4Rec | 0.0019 | 0.0017 | 0.0026 | 0.0020 | 0.0090 | 0.0058 | 0.0090 | 0.0072 | 0.0087 | 0.0073 | 0.0096 | 0.0081 |
| SASRec | 0.0042 | 0.0028 | 0.0053 | 0.0032 | 0.0098 | 0.0069 | 0.0156 | 0.0088 | 0.0108 | 0.0088 | 0.0165 | 0.0107 |
| BERT4Rec | 0.0101 | 0.0051 | 0.0149 | 0.0073 | 0.0152 | 0.0105 | 0.0266 | 0.0141 | 0.0221 | 0.0124 | 0.0308 | 0.0165 |
| CL4SRec | 0.0112 | 0.0075 | 0.0151 | 0.0085 | 0.0226 | 0.0119 | 0.0327 | 0.0169 | 0.0227 | 0.0145 | 0.0330 | 0.0170 |
| TIGER | 0.0091 | 0.0065 | 0.0160 | 0.0091 | 0.0247 | 0.0167 | 0.0378 | 0.0195 | 0.0187 | 0.0129 | 0.0251 | 0.0158 |
| DiffuRec | 0.0093 | 0.0075 | 0.0120 | 0.0083 | 0.0284 | 0.0206 | 0.0320 | 0.0226 | 0.0310 | 0.0246 | 0.0332 | 0.0251 |
| DreamRec | 0.0147 | 0.0132 | 0.0207 | 0.0134 | 0.0387 | 0.0278 | 0.0481 | 0.0314 | 0.0425 | 0.0315 | 0.0476 | 0.0342 |
| PreferDiff | 0.0188 | 0.0148 | 0.0222 | 0.0159 | 0.0420 | 0.0307 | 0.0509 | 0.0336 | 0.0453 | 0.0347 | 0.0525 | 0.0370 |
| **SteerRec** | **0.0208** | **0.0167** | **0.0275** | **0.0189** | **0.0443** | **0.0334** | **0.0531** | **0.0365** | **0.0473** | **0.0370** | **0.0592** | **0.0404** |
| Improv. (%) | +10.64% | +12.84% | +23.87% | +18.87% | +5.48% | +8.79% | +4.32% | +8.04% | +4.42% | +6.63% | +12.8% | +9.19% |

improvements are particularly pronounced on top-10 metrics such as Recall@10 and NDCG@10. For instance, on the Sports and Outdoors dataset, SteerRec achieves relative gains of 23.87% on R@10 and 18.87% on N@10 over the strongest baseline.

The superiority of SteerRec stems from its novel approach to leveraging negative signals. Unlike models such as DiffuRec and DreamRec that rely solely on positive signals, SteerRec incorporates direct negative guidance to explicitly sculpt the user's preference space by repelling the generation from undesirable items. Compared to PreferDiff, which indirectly introduces negative signals by aligning the diffusion model with a BPR loss during training, SteerRec's approach is more direct and powerful. Our GAL explicitly enables the PNG mechanism at inference time, allowing the model to use negative information to manifestly guide the denoising process. This direct, end-to-end alignment allows SteerRec to generate an ideal item embedding that is simultaneously close to user preferences and far from disliked items.

### 4.3 ABLATION STUDY (RQ2)

We designed an ablation study with two model variants to validate the contributions of SteerRec's two primary components: the PNG inference mechanism and the GAL training objective. The variants are defined as:

- **SteerRec (w/o PNG)**: This variant is trained with the full objective, including the $L_{\text{GAL}}$, but reverts to standard CFG for inference. This tests the impact of our PNG mechanism.

- **SteerRec (w/o GAL)**: This variant is trained using only the reconstruction objective but still applies PNG at inference. This tests the necessity of the alignment loss for the guidance to be effective.

Table 2: Ablation study of SteerRec's core components. The performance drop in both variants demonstrates that the PNG mechanism and the GAL are both essential.

| Model Variant | Sports and Outdoors | | | | Beauty | | | | Toys and Games | | | |
|---|---|---|---|---|---|---|---|---|---|---|---|---|
| | R@5 | N@5 | R@10 | N@10 | R@5 | N@5 | R@10 | N@10 | R@5 | N@5 | R@10 | N@10 |
| **SteerRec** | **0.0208** | **0.0167** | **0.0275** | **0.0189** | **0.0443** | **0.0334** | **0.0531** | **0.0365** | **0.0473** | **0.0370** | **0.0592** | **0.0404** |
| w/o PNG | 0.0193 | 0.0157 | 0.0244 | 0.0172 | 0.0393 | 0.0313 | 0.0474 | 0.0339 | 0.0445 | 0.0348 | 0.0539 | 0.0375 |
| w/o GAL | 0.0181 | 0.0142 | 0.0215 | 0.0153 | 0.0411 | 0.0324 | 0.0496 | 0.0351 | 0.0449 | 0.0351 | 0.0494 | 0.0367 |

The results in Table 2 indicate that both components are critical to SteerRec's performance. Removing the PNG mechanism (w/o PNG) results in a consistent performance drop, confirming that the inference-time negative guidance is a key driver of the improvements. A more substantial decline is observed for the w/o GAL variant. This underscores the necessity of the alignment loss; without it, the guidance mechanism operates on unaligned representations and becomes largely ineffective. The success of SteerRec thus stems from the synergy between its training objective and guidance paradigm.

### 4.4 HYPERPARAMETER AND EFFICIENCY ANALYSIS (RQ3)

**Impact of Guidance Scale $w$.**   We investigate the impact of the guidance scale $w$, which controls the overall guidance intensity, balancing preference fidelity against generative diversity. As shown in Figure 3, SteerRec's performance is sensitive to this value. Across all datasets, performance generally improves with a moderate increase in $w$ before declining. This demonstrates the typical trade-off where overly strong guidance can improve fidelity but narrow the generation space excessively, harming diversity. Additional analyses on other key hyperparameters, such as the loss balancing coefficient $\mu$, the triplet margin $m$, and the number of negative samples $N_{neg}$, are provided in Appendix D.4.

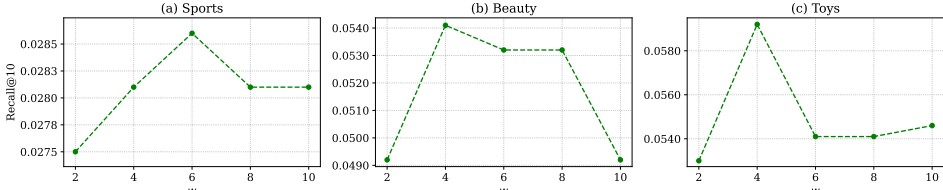

Figure 3: The impact of the guidance scale $w$ on Recall@10 performance.

**Faster Convergence than PreferDiff.**   To assess practical advantages, we analyze the training efficiency of SteerRec against PreferDiff. As shown in Figure 4, SteerRec converges significantly faster. It reaches its peak performance on the validation set around epoch 20 (approx. 6 minutes), whereas PreferDiff requires nearly twice as many iterations, converging around epoch 34 (approx. 11 minutes). This accelerated convergence stems from the direct and efficient learning signal provided by our GAL objective. By immediately training the model to distinguish between positive and negative conditions, SteerRec directly aligns the training process with the PNG inference mechanism. In contrast, PreferDiff's indirect alignment of its BPR loss with the standard CFG objective results in a slower learning process. Similar convergence trends on the other datasets are provided in Appendix D.5.

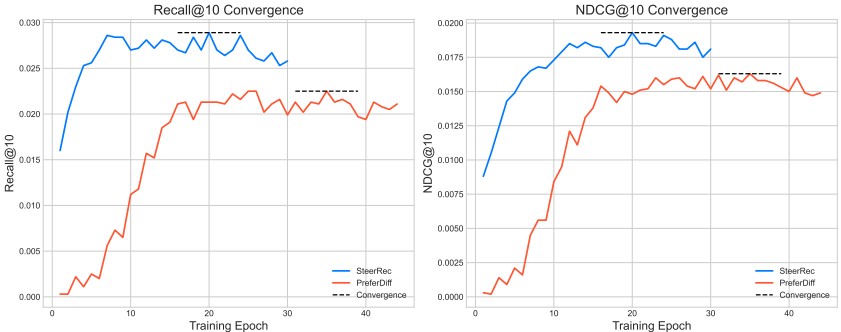

Figure 4: Training performance comparison between SteerRec and PreferDiff on the Sports dataset.

## 5 CONCLUSION AND LIMITATIONS

In this work, we introduced SteerRec, a framework that redefines guidance for diffusion-based recommenders. Instead of the standard CFG, SteerRec replaces the generic unconditional prior with a user-aware negative condition to enable more precise, personalized repulsion. To facilitate this, we proposed the GAL, an alignment loss that ensures the model can distinguish between positive and negative guidance, leading to significant performance gains as validated by our experiments. Key limitations of our framework include its dependence on the quality of negative samples and the potential for designing more advanced alignment loss functions. Future work could also explore applying negative guidance primarily in the later stages of the denoising process, drawing inspiration from similar techniques in image generation.

## ETHICS STATEMENT

The research presented in this paper focuses on algorithmic advancements for sequential recommendation. We exclusively used publicly available, anonymized benchmark datasets (Amazon Reviews), which are standard in the academic community for recommendation research. Our work does not involve collecting new data from human subjects, nor does it deal with sensitive personal information. The proposed method, SteerRec, aims to improve recommendation accuracy and does not introduce inherent fairness or privacy risks beyond those generally associated with recommender systems. We believe our work adheres to the ICLR Code of Ethics.

## REPRODUCIBILITY STATEMENT

To ensure the reproducibility of our work, we have made our implementation publicly available at the anonymous URL provided on the first page. All experiments were conducted on three well-known public datasets: Sports and Outdoors, Beauty, and Toys and Games from the Amazon Review collection. Comprehensive details regarding the model architecture, training procedures, and hyper-parameter settings for all experiments are provided in Appendix D.2 and D.3. Our implementation is available at https://anonymous.4open.science/r/SteerRec-5D70.

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

## A    USE OF LLMS

During the preparation of this manuscript, we utilized a large language model (LLM) as a writing assistant. Its role was primarily to aid in polishing and refining prose, improving grammar, and ensuring clarity and conciseness in our descriptions. The LLM was not used for research ideation, conducting experiments, or generating core theoretical and methodological contributions presented in this work. All final content, including the scientific claims and technical details, was written and verified by the authors, who take full responsibility for the paper.

## B    RELATED WORK

**Sequential Recommendation.**    The task of sequential recommendation has evolved significantly. Early approaches often relied on Markov chains to model item-to-item transitions (Rendle et al., 2010). The field saw a major shift with the introduction of recurrent neural networks (RNNs) to capture the temporal dynamics of user sequences (Hidasi et al., 2016). Subsequently, Transformer-based models, leveraging self-attention mechanisms, became the standard due to their superior ability to capture long-range dependencies in user behavior (Kang & McAuley, 2018; Sun et al., 2019). More recently, to address data sparsity and improve representation robustness, self-supervised learning, particularly contrastive learning, has been successfully applied (Zhou et al., 2020; Xie et al., 2022). Our work builds upon this rich history but explores the problem from a novel generative, rather than discriminative, perspective.

**DMs in Recommendation.**    The success of DMs in high-fidelity data synthesis (Sohl-Dickstein et al., 2015; Ho et al., 2020; Karras et al., 2022; Song et al., 2021b) has led to their broad application across numerous fields, including image (Rombach et al., 2022; Saharia et al., 2022; Ramesh et al., 2022; Dhariwal & Nichol, 2021), video (Brooks et al., 2024; Ho et al., 2022), audio (Kong et al., 2021; Popov et al., 2021), 3D modeling (Poole et al., 2023), natural language (Li et al., 2022), and molecular biology (Hoogeboom et al., 2022; Corso et al., 2023). This proven ability to model complex distributions has motivated their exploration in recommender systems. The application of DMs to recommendation has primarily evolved along two technical pathways. The first approach frames recommendation as a generative profile completion task. Models like DiffRec and CF-Diff treat a user's entire multi-hot interaction vector as the data to be diffused, learning to restore the full profile from a noisy version (Wang et al., 2023b; Hou et al., 2024). A second, distinct pathway employs an explicitly conditional generation framework, which is prevalent in sequential recommendation. In these models, a user's chronological interaction history is encoded into a context vector to guide the generation of a single next-item embedding (Yang et al., 2023; Li et al., 2023; Wang et al., 2024). A common thread uniting these conditional models is their reliance on positive signals for guidance. PreferDiff (Liu et al., 2025) made an important step forward by incorporating negative information, adding a BPR-inspired loss to the training objective to learn more discriminative representations. However, a critical gap remains. The use of user-aware negative signals to directly steer the inference-time guidance process is an unexplored area. All prior conditional diffusion recommenders, including PreferDiff, still rely on the standard user-agnostic CFG for inference. SteerRec is the first framework designed to fill this gap, introducing a novel paradigm where negative feedback directly shapes the generative process at inference time.

## C    THEORETICAL JUSTIFICATION OF THE STEERREC FRAMEWORK

This appendix provides a comprehensive theoretical justification for the SteerRec framework, starting from first principles. We begin by tracing the mathematical evolution of guidance mechanisms in diffusion models, establishing the context and motivation for our approach. We then present a detailed derivation of our PNG mechanism, connecting its score-based formulation to the practical direct $x_0$ prediction used in our implementation. Finally, we provide a rigorous analysis of our novel GAL, tracing its origins in deep metric learning and proving its alignment with our inference objective.

### C.1 THE EVOLUTION OF GUIDANCE IN DIFFUSION MODELS

Guidance mechanisms are central to making DMs controllable generative tools. Our work builds upon and extends a rich lineage of guidance techniques, which we detail below.

**Classifier Guidance (CG).** The original concept of guidance, introduced by (Dhariwal & Nichol, 2021), leverages a separately trained classifier $p_\phi(c|\mathbf{x}_t)$ to steer the generation process. The core idea is to modify the score of the unconditional distribution $\nabla_{\mathbf{x}_t} \log p(\mathbf{x}_t)$ by adding the gradient of the log-likelihood from the classifier. This is formally derived from Bayes' rule:

$$p(\mathbf{x}_t|c) = \frac{p(c|\mathbf{x}_t)p(\mathbf{x}_t)}{p(c)} \tag{15}$$

By taking the logarithm and then the gradient with respect to the noisy data $\mathbf{x}_t$, we obtain the score function of the conditional distribution:

$$\log p(\mathbf{x}_t|c) = \log p(c|\mathbf{x}_t) + \log p(\mathbf{x}_t) - \log p(c) \tag{16}$$
$$\nabla_{\mathbf{x}_t} \log p(\mathbf{x}_t|c) = \nabla_{\mathbf{x}_t} \log p(\mathbf{x}_t) + \nabla_{\mathbf{x}_t} \log p_\phi(c|\mathbf{x}_t) \tag{17}$$

Here, the term $s_\theta(\mathbf{x}_t) = \nabla_{\mathbf{x}_t} \log p(\mathbf{x}_t)$ represents the score of the unconditional diffusion model, and $\nabla_{\mathbf{x}_t} \log p_\phi(c|\mathbf{x}_t)$ is the gradient provided by the external classifier. The relationship between the score function and the noise prediction $\epsilon_\theta$ of a diffusion model is given by $s_\theta(\mathbf{x}_t) = -\epsilon_\theta(\mathbf{x}_t)/\sqrt{1 - \bar{\alpha}_t}$. By substituting this relationship, we can derive the guided noise prediction $\hat{\epsilon}_\theta(\mathbf{x}_t, c)$:

$$\hat{\epsilon}_\theta(\mathbf{x}_t, c) = \epsilon_\theta(\mathbf{x}_t) - w\sqrt{1 - \bar{\alpha}_t} \cdot \nabla_{\mathbf{x}_t} \log p_\phi(c|\mathbf{x}_t) \tag{18}$$

where $w$ is the guidance scale. While powerful, this approach requires training a separate classifier on noisy data, which adds significant complexity and computational overhead.

**CFG.** To overcome the limitations of CG, (Ho & Salimans, 2022) proposed CFG. The core insight is to train a single conditional model $\epsilon_\theta(\mathbf{x}_t, c)$ to also operate unconditionally by randomly replacing the condition $c$ with a null token $\emptyset$ during training. At inference, the guidance is formulated as an extrapolation away from the unconditional prediction:

$$\hat{\epsilon}_\theta(\mathbf{x}_t, c) = \epsilon_\theta(\mathbf{x}_t, \emptyset) + w \cdot (\epsilon_\theta(\mathbf{x}_t, c) - \epsilon_\theta(\mathbf{x}_t, \emptyset)) \tag{19}$$

The term $(\epsilon_\theta(\mathbf{x}_t, c) - \epsilon_\theta(\mathbf{x}_t, \emptyset))$ can be seen as an approximation of the classifier gradient from CG. We can formalize this connection. Consider an implicit classifier $p(c|\mathbf{x}_t) \propto p(\mathbf{x}_t|c)/p(\mathbf{x}_t)$. Its log-gradient is:

$$\nabla_{\mathbf{x}_t} \log p(c|\mathbf{x}_t) = \nabla_{\mathbf{x}_t} \log p(\mathbf{x}_t|c) - \nabla_{\mathbf{x}_t} \log p(\mathbf{x}_t)$$
$$\approx -\frac{1}{\sqrt{1 - \bar{\alpha}_t}} (\epsilon_\theta(\mathbf{x}_t, c) - \epsilon_\theta(\mathbf{x}_t, \emptyset)) \tag{20}$$

Substituting this implicit gradient back into the original CG formula gives a result that closely resembles the CFG rule, demonstrating that CFG is a principled and efficient approximation of CG.

### C.2 DETAILED DERIVATION OF THE STEERREC PNG MECHANISM

PNG evolves from CFG, replacing the "positive-vs-unconditional" structure with a more powerful "positive-vs-negative" paradigm, designed for optimal discrimination between two opposing user preferences.

**Guidance via Optimal Discrimination.** The philosophy of CG and CFG is one of Bayesian estimation. Our SteerRec framework, however, addresses the different goal of actively discriminating between a positive condition $\mathbf{c}^+$ and a negative condition $\mathbf{c}^-$. The optimal statistic for such a task is the likelihood ratio, as established by the Neyman-Pearson Lemma (Neyman & Pearson, 1933). We therefore posit that a principled generative process should be guided by a score function that reflects this statistic at each denoising step $t$. This is achieved by defining a target marginal distribution at

each time $t$, $p_\theta^*(\mathbf{x}_t|\cdot)$, to be directly proportional to the likelihood ratio, sharpened by a guidance scale $w$.

The SteerRec guidance mechanism is derived from a target probability distribution $p_\theta^*$ defined by the conditional likelihood ratio:

$$p_\theta^*(\mathbf{x}_t|\mathbf{c}^+, \mathbf{c}^-) \propto \frac{p_\theta(\mathbf{x}_t|\mathbf{c}^+)^{1+w}}{p_\theta(\mathbf{x}_t|\mathbf{c}^-)^w} \tag{21}$$

The proof proceeds by deriving the score function of the posited target distribution. We begin with the logarithm of the distribution in Eq. 21:

$$\log p_\theta^*(\mathbf{x}_t|\mathbf{c}^+, \mathbf{c}^-) = (1+w)\log p_\theta(\mathbf{x}_t|\mathbf{c}^+) - w\log p_\theta(\mathbf{x}_t|\mathbf{c}^-) - \log Z \tag{22}$$

where $Z$ is the partition function. Applying the gradient operator $\nabla_{\mathbf{x}_t}$ to derive the score function $s_\theta^* := \nabla_{\mathbf{x}_t} \log p_\theta^*$:

$$s_\theta^*(\mathbf{x}_t, \mathbf{c}^+, \mathbf{c}^-) = \nabla_{\mathbf{x}_t}\left[(1+w)\log p_\theta(\mathbf{x}_t|\mathbf{c}^+) - w\log p_\theta(\mathbf{x}_t|\mathbf{c}^-) - \log Z\right] \tag{23}$$

$$= (1+w)\nabla_{\mathbf{x}_t}\log p_\theta(\mathbf{x}_t|\mathbf{c}^+) - w\nabla_{\mathbf{x}_t}\log p_\theta(\mathbf{x}_t|\mathbf{c}^-) \quad (\text{since } \nabla_{\mathbf{x}_t}\log Z = 0) \tag{24}$$

$$= (1+w)s_\theta(\mathbf{x}_t, \mathbf{c}^+, t) - w \cdot s_\theta(\mathbf{x}_t, \mathbf{c}^-, t) \tag{25}$$

The final derived score function (Eq. 25) is the score-space formulation of our guidance rule.

**From Score Functions to Direct $\mathbf{x}_0$ Prediction.** While theoretically grounded in score matching, our model $F_\theta(\mathbf{x}_t, \mathbf{c}, t)$ is parameterized to directly predict the clean data $\mathbf{x}_0$. The score function $s_\theta$ and the predicted $\mathbf{x}_0$ are intrinsically linked via the relationship $s_\theta(\mathbf{x}_t, \mathbf{c}, t) = -(\mathbf{x}_t - \sqrt{\bar\alpha_t}F_\theta(\mathbf{x}_t, \mathbf{c}, t))/(1 - \bar\alpha_t)$. We now substitute this relationship back into our derived score guidance rule (Eq. 25). Let $\hat{\mathbf{x}}_0^{\text{pos}} = F_\theta(\mathbf{x}_t, \mathbf{c}^+, t)$ and $\hat{\mathbf{x}}_0^{\text{neg}} = F_\theta(\mathbf{x}_t, \mathbf{c}^-, t)$. This yields a guided score $s_\theta^*$:

$$s_\theta^*(\mathbf{x}_t, \dots) = (1+w)\left(-\frac{\mathbf{x}_t - \sqrt{\bar\alpha_t}\hat{\mathbf{x}}_0^{\text{pos}}}{1 - \bar\alpha_t}\right) - w\left(-\frac{\mathbf{x}_t - \sqrt{\bar\alpha_t}\hat{\mathbf{x}}_0^{\text{neg}}}{1 - \bar\alpha_t}\right)$$

$$= \frac{1}{1 - \bar\alpha_t}\left[-(1+w)(\mathbf{x}_t - \sqrt{\bar\alpha_t}\hat{\mathbf{x}}_0^{\text{pos}}) + w(\mathbf{x}_t - \sqrt{\bar\alpha_t}\hat{\mathbf{x}}_0^{\text{neg}})\right]$$

$$= \frac{1}{1 - \bar\alpha_t}\left[(-1 - w + w)\mathbf{x}_t + (1+w)\sqrt{\bar\alpha_t}\hat{\mathbf{x}}_0^{\text{pos}} - w\sqrt{\bar\alpha_t}\hat{\mathbf{x}}_0^{\text{neg}}\right]$$

$$= -\frac{\mathbf{x}_t - \sqrt{\bar\alpha_t}\left((1+w)\hat{\mathbf{x}}_0^{\text{pos}} - w\hat{\mathbf{x}}_0^{\text{neg}}\right)}{1 - \bar\alpha_t} \tag{26}$$

By comparing this result back to the structure of the score-to-$\mathbf{x}_0$ relationship, we can identify the term in the parenthesis as the guided prediction of the clean data, $\hat{\mathbf{x}}_0$. Therefore, we arrive at the final guidance rule in the $\mathbf{x}_0$ prediction space:

$$\hat{\mathbf{x}}_0(\mathbf{x}_t, \mathbf{c}^+, \mathbf{c}^-) = (1+w)F_\theta(\mathbf{x}_t, \mathbf{c}^+, t) - wF_\theta(\mathbf{x}_t, \mathbf{c}^-, t) \tag{27}$$

This detailed derivation confirms that the intuitive guidance formula used in our implementation is a direct and principled consequence of the score-based formulation.

## C.3 THEORETICAL ANALYSIS OF THE GAL

The effectiveness of the PNG mechanism hinges on the model's ability to produce semantically distinct outputs under opposing conditions. We designed the GAL to explicitly instill this capability.

**Motivation from Deep Metric Learning.** Our loss function is directly inspired by the Triplet Loss, a cornerstone of deep metric learning popularized by (Schroff et al., 2015) for face recognition. The goal of metric learning is to learn an embedding space where similar inputs are mapped to nearby points and dissimilar inputs are mapped to distant points. For an anchor sample $a$, a positive

sample $p$ (of the same identity as $a$), and a negative sample $n$ (of a different identity), the Triplet Loss is formulated as:

$$L_{\text{triplet}} = \sum_{i}^{N} \left[ \|\mathbf{f}(\mathbf{x}_i^a) - \mathbf{f}(\mathbf{x}_i^p)\|_2^2 - \|\mathbf{f}(\mathbf{x}_i^a) - \mathbf{f}(\mathbf{x}_i^n)\|_2^2 + m \right]_+ \tag{28}$$

where $\mathbf{f}(\cdot)$ is the embedding function, $m$ is a margin, and $[z]_+ = \max(0, z)$. This loss penalizes the model unless the distance between the anchor and the positive is smaller than the distance between the anchor and the negative by at least the margin $m$.

We adapt this powerful principle to our generative context. Instead of operating on static input embeddings, we apply the triplet constraint to the dynamic outputs of our denoising network under different conditions. The analogy is as follows:

- **Anchor ($a$):** The ground-truth item embedding, $\mathbf{x}_0^+$.
- **Positive ($p$):** The denoised prediction under positive guidance, $\hat{\mathbf{x}}_0^{\text{pos}} = F_\theta(\mathbf{x}_t, \mathbf{c}^+, t)$.
- **Negative ($n$):** The denoised prediction under negative guidance, $\hat{\mathbf{x}}_0^{\text{neg}} = F_\theta(\mathbf{x}_t, \mathbf{c}^-, t)$.

Our $L_{\text{GAL}}$ directly instantiates this logic, ensuring that the generative process itself learns to respect the desired preference structure.

**Gradient-based Proof of Alignment.** Minimizing the $L$ objective directly optimizes for the fidelity and separability required by the inference mechanism.

We analyze the gradient of the total loss with respect to the model parameters $\theta$, $\nabla_\theta L$. The total loss is:

$$L = (1 - \mu) \cdot d(\hat{\mathbf{x}}_0^{\text{pos}}, \mathbf{x}_0^+) + \mu \cdot \max(0, d(\hat{\mathbf{x}}_0^{\text{pos}}, \mathbf{x}_0^+) - d(\hat{\mathbf{x}}_0^{\text{neg}}, \mathbf{x}_0^+) + m) \tag{29}$$

When the margin constraint is violated (i.e., when the term inside $\max(0, \dots)$ is positive), the gradient of the GAL part is non-zero. The total gradient becomes:

$$\nabla_\theta L = (1 - \mu)\nabla_\theta d(\hat{\mathbf{x}}_0^{\text{pos}}, \mathbf{x}_0^+) + \mu \left( \nabla_\theta d(\hat{\mathbf{x}}_0^{\text{pos}}, \mathbf{x}_0^+) - \nabla_\theta d(\hat{\mathbf{x}}_0^{\text{neg}}, \mathbf{x}_0^+) \right) \tag{30}$$

$$= (1 - \mu + \mu)\nabla_\theta d(\hat{\mathbf{x}}_0^{\text{pos}}, \mathbf{x}_0^+) - \mu\nabla_\theta d(\hat{\mathbf{x}}_0^{\text{neg}}, \mathbf{x}_0^+) \tag{31}$$

$$= \nabla_\theta d(\hat{\mathbf{x}}_0^{\text{pos}}, \mathbf{x}_0^+) - \mu\nabla_\theta d(\hat{\mathbf{x}}_0^{\text{neg}}, \mathbf{x}_0^+) \tag{32}$$

This gradient consists of two opposing forces acting on the model parameters:

1. The term $\nabla_\theta d(\hat{\mathbf{x}}_0^{\text{pos}}, \mathbf{x}_0^+)$ forces the model to adjust its parameters to make its positive prediction $\hat{\mathbf{x}}_0^{\text{pos}}$ closer to the ground truth $\mathbf{x}_0^+$. This directly optimizes for fidelity.

2. The term $-\mu\nabla_\theta d(\hat{\mathbf{x}}_0^{\text{neg}}, \mathbf{x}_0^+)$ forces the model to adjust its parameters to make its negative prediction $\hat{\mathbf{x}}_0^{\text{neg}}$ further away from the ground truth $\mathbf{x}_0^+$. This directly optimizes for separability, creating the semantically distinct outputs that the inference rule relies on.

When the margin is satisfied, $L_{\text{GAL}} = 0$ and the gradient is simply $(1 - \mu)\nabla_\theta d(\hat{\mathbf{x}}_0^{\text{pos}}, \mathbf{x}_0^+)$, focusing solely on improving reconstruction. The training objective thus dynamically supplies precisely the two properties demanded by the inference rule, establishing a tight alignment by design.

## C.4 CONNECTION TO THE DDIM SAMPLING PROCESS

The final generated item embedding is produced by the DDIM sampler, which iteratively uses the guided prediction $\hat{\mathbf{x}}_0$. The one-step update from a noisy state $\mathbf{x}_t$ to a less noisy state $\mathbf{x}_{t-1}$ is given by:

$$\mathbf{x}_{t-1} = \sqrt{\bar{\alpha}_{t-1}}\hat{\mathbf{x}}_0 + \sqrt{1 - \bar{\alpha}_{t-1}} \cdot \hat{\epsilon}_\theta(\mathbf{x}_t, \hat{\mathbf{x}}_0) \tag{33}$$

where the predicted noise $\hat{\epsilon}_\theta$ is derived from the guided prediction $\hat{\mathbf{x}}_0$:

$$\hat{\epsilon}_\theta(\mathbf{x}_t, \hat{\mathbf{x}}_0) = \frac{\mathbf{x}_t - \sqrt{\bar{\alpha}_t}\hat{\mathbf{x}}_0}{\sqrt{1 - \bar{\alpha}_t}} \tag{34}$$

Here, $\hat{\mathbf{x}}_0$ is exactly the output of our SteerRec guidance rule (Eq. 27). This completes the chain, showing how our principled guidance mechanism integrates seamlessly into the established DDIM sampling process to generate the final preference item embedding.

## D EXPERIMENTAL SETUP

This section provides a comprehensive overview of our experimental setup to ensure full reproducibility.

**Datasets and Preprocessing.** We conduct experiments on three public benchmark datasets from the Amazon Review 2014 collection[1]: Sports and Outdoors, Beauty, and Toys and Games. To ensure a direct and fair comparison with recent work, we utilize the identical preprocessed data and user-based data splits (80% train, 10% validation, 10% test) as publicly released by (Liu et al., 2025). The protocol involves five-core filtering, where users and items with fewer than five interactions are iteratively removed. For sequence construction, we use the last 10 interactions as the input context for predicting the next item. Sequences with fewer than 10 interactions are post-padded with a special padding token (ID 0). The key statistics of the datasets after preprocessing are summarized in Table 3.

Table 3: Detailed statistics of the datasets after preprocessing.

| Dataset | #Sequences | #Items | #Interactions |
|---|---|---|---|
| Sports and Outdoors | 35,598 | 18,357 | 256,598 |
| Beauty | 22,363 | 12,101 | 162,150 |
| Toys and Games | 19,412 | 11,924 | 138,444 |

### D.1 BASELINE MODEL DESCRIPTIONS

We compare SteerRec against a comprehensive suite of baseline models, categorized as follows.

**Traditional Sequential Models.**

- **GRU4Rec** (Hidasi et al., 2016): Employs Gated Recurrent Units (GRUs) to model the temporal dynamics within user interaction sequences for session-based recommendation.
- **SASRec** (Kang & McAuley, 2018): A seminal work that introduced the Transformer architecture with causal self-attention to capture item-item transitions for recommendation.
- **BERT4Rec** (Sun et al., 2019): Adapts the bidirectional Transformer architecture from NLP, using a Cloze (masked item prediction) objective to learn deep sequential representations.

**Contrastive and Generative Models.**

- **CL4SRec** (Xie et al., 2022): Augments a Transformer-based model with a contrastive learning objective, learning robust sequence representations by maximizing agreement between different augmented views of the same sequence.
- **TIGER** (Rajput et al., 2023): A generative model that reframes recommendation as a sequence-to-sequence task by quantizing item semantics into discrete codes using a VQ-VAE, which are then predicted autoregressively.

**Diffusion-based Models.**

- **DiffuRec** (Li et al., 2023): A diffusion model for sequential recommendation that is trained with a standard cross-entropy loss, where the noised target item embedding is used to modulate historical item representations.
- **DreamRec** (Yang et al., 2023): A foundational diffusion-based recommender that generates a next-item embedding guided by the user's history. It is trained with a reconstruction loss and uses standard CFG at inference.

---

[1] https://cseweb.ucsd.edu/~jmcauley/datasets/amazon/links.html

- **PreferDiff**[2] (Liu et al., 2025): An enhancement over DreamRec that incorporates a BPR-style preference loss during training to learn from negative samples. It still relies on the standard user-agnostic CFG for inference.

## D.2 IMPLEMENTATION DETAILS

**Environment and Reproducibility.** All experiments were implemented in PyTorch and executed on a single NVIDIA V100-SXM2 GPU with 32GB memory. We set a fixed random seed for all stochastic operations to ensure reproducibility.

**Shared Model Architecture.** To ensure a fair comparison and isolate the benefits of our framework, SteerRec is built upon the same SASRec backbone[3] as DreamRec and PreferDiff. This backbone consists of a single Transformer layer with two attention heads. Consistent with these baselines, the item embedding dimension for all diffusion-based models is set to 3072. Following PreferDiff, the denoising network is implemented as a simple linear projection layer. All model parameters are initialized using a standard normal distribution.

**Training Details.** We use the AdamW optimizer for training all models. For the loss computations in our framework, we use Cosine distance as the distance metric $d(\cdot, \cdot)$. This choice follows PreferDiff (Liu et al., 2025), which demonstrated that Cosine distance is particularly effective for recommendation tasks. We employ an early stopping strategy with a patience of 10 epochs, monitored on the validation set's Recall@5 performance. The training batch size is set to 256 for all experiments.

---

**Algorithm 1** Training Procedure of SteerRec

1: **repeat**
2: $(\mathbf{c}^+, \mathbf{x}_0^+) \sim \mathcal{D}$ ▷ Sample a context (user history) and its target item
3: $\mathbf{c}^- \leftarrow$ Construct Negative Condition ▷ Construct the negative condition for training
4: $t \sim \text{Uniform}(\{1, \ldots, T\})$ ▷ Sample a random timestep
5: $\epsilon \sim \mathcal{N}(\mathbf{0}, \mathbf{I})$ ▷ Sample a Gaussian noise vector
6: $\mathbf{x}_t \leftarrow \sqrt{\bar{\alpha}_t}\mathbf{x}_0^+ + \sqrt{1 - \bar{\alpha}_t}\epsilon$ ▷ Corrupt the target item via the forward process
7: $\hat{\mathbf{x}}_0^{\text{pos}} \leftarrow F_\theta(\mathbf{x}_t, \mathbf{c}^+, t)$ ▷ Denoise using the positive condition
8: $\hat{\mathbf{x}}_0^{\text{neg}} \leftarrow F_\theta(\mathbf{x}_t, \mathbf{c}^-, t)$ ▷ Denoise using the negative condition
9: $L_{\text{recon}} \leftarrow d(\hat{\mathbf{x}}_0^{\text{pos}}, \mathbf{x}_0^+)$ ▷ Compute the reconstruction loss
10: $L_{\text{GAL}} \leftarrow \max(0, d(\hat{\mathbf{x}}_0^{\text{pos}}, \mathbf{x}_0^+) - d(\hat{\mathbf{x}}_0^{\text{neg}}, \mathbf{x}_0^+) + m)$ ▷ Compute the GAL
11: $L \leftarrow (1 - \mu) \cdot L_{\text{recon}} + \mu \cdot L_{\text{GAL}}$ ▷ Compute the final composite loss
12: Take a gradient descent step on $\nabla_\theta L$ ▷ Update model parameters
13: **until** converged

---

**Algorithm 2** Inference Procedure of SteerRec

**Require:** User history context $\mathbf{c}^+$, guidance scale $w$
1: $\mathbf{c}^- \leftarrow$ Construct Negative Condition ▷ Construct the negative condition for inference
2: $\mathbf{x}_T \sim \mathcal{N}(\mathbf{0}, \mathbf{I})$ ▷ Sample initial noise from the prior distribution
3: **for** $t = T, \ldots, 1$ **do** ▷ Begin the reverse denoising loop
4: $\hat{\mathbf{x}}_0^{\text{pos}} \leftarrow F_\theta(\mathbf{x}_t, \mathbf{c}^+, t)$ ▷ Predict clean item $\mathbf{x}_0$ with the positive condition
5: $\hat{\mathbf{x}}_0^{\text{neg}} \leftarrow F_\theta(\mathbf{x}_t, \mathbf{c}^-, t)$ ▷ Predict clean item $\mathbf{x}_0$ with the negative condition
6: $\hat{\mathbf{x}}_0 \leftarrow (1 + w)\hat{\mathbf{x}}_0^{\text{pos}} - w \cdot \hat{\mathbf{x}}_0^{\text{neg}}$ ▷ Apply PNG mechanism to get the guided prediction
7: $\hat{\epsilon}_\theta \leftarrow (\mathbf{x}_t - \sqrt{\bar{\alpha}_t}\hat{\mathbf{x}}_0)/\sqrt{1 - \bar{\alpha}_t}$ ▷ Estimate the corresponding noise $\epsilon$
8: $\mathbf{x}_{t-1} \leftarrow \sqrt{\bar{\alpha}_{t-1}}\hat{\mathbf{x}}_0 + \sqrt{1 - \bar{\alpha}_{t-1}} \cdot \hat{\epsilon}_\theta$ ▷ Perform one DDIM update step
9: **end for**
10: **return** $\hat{\mathbf{x}}_0$ ▷ Return the final denoised item embedding

---

[2] https://github.com/lswhim/PreferDiff
[3] https://github.com/kang205/SASRec

**Diffusion-Specific Parameters.** We use a linear noise schedule for $\beta_t$ over $T$ total timesteps, with $\beta_{start} = 10^{-4}$ and $\beta_{end} = 0.02$. During inference, we use the deterministic DDIM sampler (Song et al., 2021a) for efficient generation. SteerRec often achieves optimal performance with a smaller number of total timesteps $T$ (e.g., 800-1200) compared to PreferDiff that may require more (e.g., 2000-5000). We attribute this to our more direct and structured training objective. Our GAL objective imposes a complex geometric constraint at every denoising step—forcing the model to distinguish between positive and negative conditions with a margin. Learning this intricate separation at extremely fine-grained noise levels (as required by a very large $T$) can be challenging and inefficient. Instead, GAL provides such a potent and explicit signal about the relative preference structure that the model learns the desired geometry more rapidly, obviating the need for a long, fine-grained denoising chain.

### D.3 HYPERPARAMETER SETTINGS

Our framework's hyperparameters were tuned via a grid search on the validation set. The search spaces are detailed in Table 4. The best-performing configurations for SteerRec's key hyperparameters are presented in Table 5.

Table 4: Hyperparameter search space for SteerRec.

| Hyperparameter | Search Space |
|---|---|
| Learning Rate (lr) | $\{2 \cdot 10^{-4}, 1 \cdot 10^{-4}, 5 \cdot 10^{-5}, 1 \cdot 10^{-5}\}$ |
| Guidance Scale ($w$) | $\{2, 4, 6, 8, 10\}$ |
| Loss Balance ($\mu$) | $\{0.2, 0.4, 0.6, 0.8\}$ |
| Triplet Margin ($m$) | $\{0.05, 0.1, 0.2, 0.3, 0.4\}$ |
| Diffusion Timesteps ($T$) | $\{600, 800, 1000, 1200, 1400, 1600, 1800, 2000\}$ |
| Inference Negative Samples ($K$) | $\{16, 32, 64, 128, 256\}$ |

Table 5: Best settings for SteerRec's key hyperparameters on each dataset.

| Dataset | Guidance Scale ($w$) | Loss Balance ($\mu$) | Timesteps ($T$) | Margin ($m$) |
|---|---|---|---|---|
| Sports and Outdoors | 2 | 0.4 | 1000 | 0.1 |
| Beauty | 4 | 0.2 | 1200 | 0.1 |
| Toys and Games | 4 | 0.2 | 800 | 0.1 |

### D.4 ADDITIONAL HYPERPARAMETER ANALYSIS

This section provides a detailed sensitivity analysis of the key hyperparameters introduced in our framework beyond the guidance scale $w$ presented in the main text. We analyze the impact of the loss balancing coefficient $\mu$, the triplet margin $m$, and the number of inference-time negative samples $N_{neg}$.

**Impact of Loss Balancing Coefficient $\mu$.** The coefficient $\mu$ balances the reconstruction loss and the GAL. As shown in Figure 5, its value is critical for model performance. An excessively large $\mu$ (e.g., 1.0) overemphasizes preference separation, which compromises generative quality and ultimately harms performance. The optimal performance is consistently found when $\mu$ is in the moderate range of $[0.2, 0.6]$, indicating that a balanced contribution from both objectives is crucial for SteerRec.

**Impact of Triplet Margin $m$.** The margin $m$ in the $L_{\text{GAL}}$ objective sets the desired separation between the positive and negative predictions during training. Figure 6 shows that the model is sensitive to this value. A very small margin (e.g., 0.05) may not provide a strong enough alignment signal to effectively structure the embedding space. As the margin increases to a moderate value (typically in the range of $[0.1, 0.2]$), performance improves significantly. However, an overly large

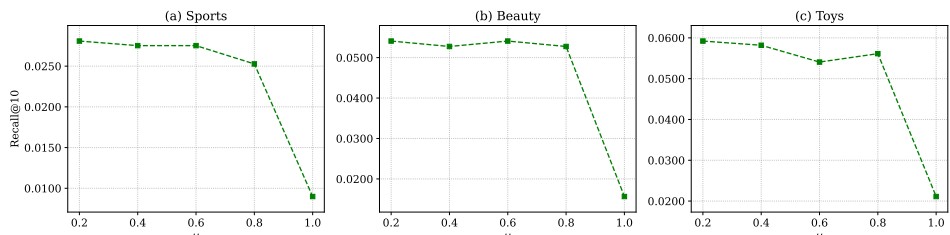

Figure 5: The impact of the loss balancing coefficient $\mu$ on recall@10 performance.

margin (e.g., 0.4) can make the training objective too difficult to satisfy, especially for hard negatives, thus hindering convergence and degrading performance. This demonstrates the importance of selecting a well-calibrated margin.

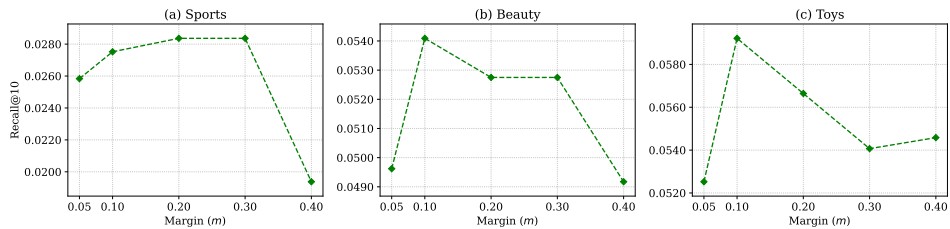

Figure 6: The impact of the triplet margin $m$ on Recall@10 performance.

**Impact of Inference Negative Samples $N_{neg}$.** The number of negative samples $N_{neg}$ used to construct the repulsive signal $\mathbf{c}^-$ at inference time also has a notable impact on performance, as shown in Figure 7. Using too few negative samples (e.g., 16) may result in an unstable or biased anti-preference signal, leading to lower performance. As $N_{neg}$ increases, the performance generally peaks with a moderate number of samples, typically around $N_{neg} = 32$ or $N_{neg} = 64$ for most datasets. This suggests that a sufficiently representative set of negatives is enough to form an effective repulsive vector. Interestingly, using an excessive number of negatives (e.g., 128 or 256) does not yield further improvements and can even slightly degrade performance, possibly due to the introduction of noisy or less relevant negative signals into the aggregated centroid.

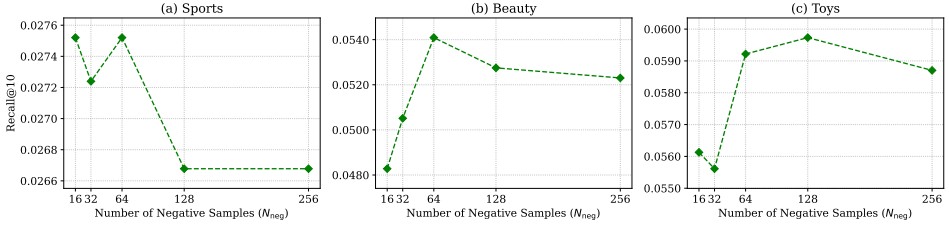

Figure 7: The impact of the number of inference-time negative samples $N_{neg}$ on Recall@10 performance.

### D.5 ADDITIONAL CONVERGENCE ANALYSIS

As established in the main paper, SteerRec demonstrates significantly accelerated convergence compared to the PreferDiff across all datasets. This appendix provides the supplementary convergence curves for the Beauty and Toys and Games datasets, which complement the results for the Sports dataset shown in the main text. Figures 8 and 9 illustrate this trend. In both cases, SteerRec (blue curve) not only converges to a higher performance plateau but also exhibits a much steeper initial

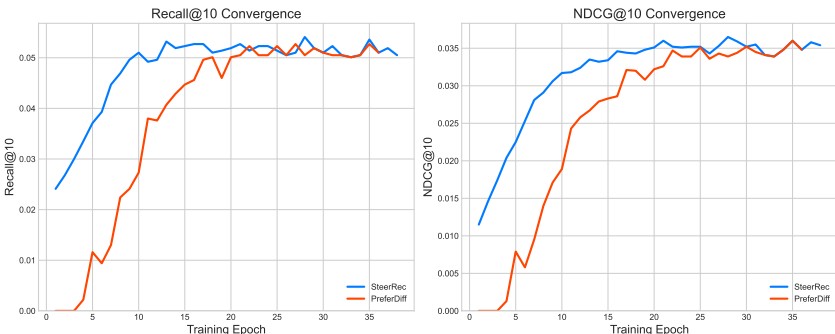

Figure 8: Training performance comparison between SteerRec and PreferDiff on the Beauty dataset.

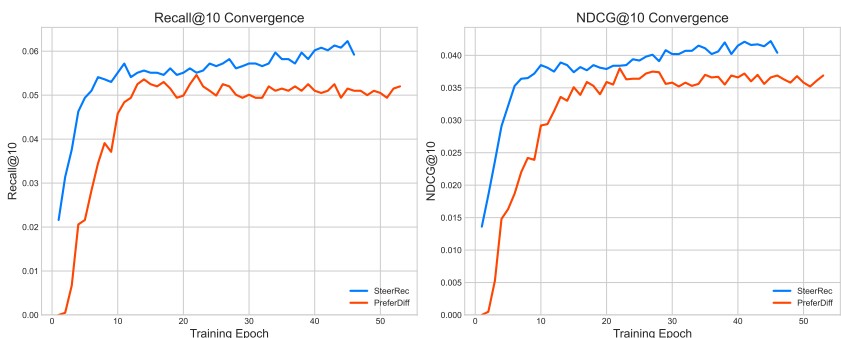

Figure 9: Training performance comparison between SteerRec and PreferDiff on the Toys and Games dataset.

learning curve, reaching near-optimal performance in substantially fewer epochs than PreferDiff (red curve).

# E  EMBEDDING SPACE VISUALIZATION

To better understand how SteerRec structures the item representation space, we visualize the learned embeddings with t-SNE[4] (Maaten & Hinton, 2008) and compare SteerRec against SASRec and PreferDiff. Figures 10, 11, and 12 show results for the Sports, Beauty, and Toys datasets. A consistent trend emerges across all datasets. SASRec learns a highly concentrated embedding space, with items crowded into a dense and largely unstructured core. PreferDiff, which incorporates a preference loss, explores the space more broadly but forms uneven clusters with indistinct boundaries. In contrast, SteerRec produces a well-structured representation space with multiple distinct and dense clusters separated by clear low-density regions.

# F  EXTENDED EXPERIMENTS ON THE MIND-SMALL DATASET

To further validate the effectiveness of SteerRec, especially in settings with rich negative feedback, we conduct experiments on the widely-used MIND-small news recommendation dataset[5] (Wu et al., 2020). This dataset is particularly suitable for our study because it provides explicit negative signals and has been widely adopted in news recommendation research (Wu et al., 2019a;b). Below we describe the dataset, experimental setup, and results.

---

[4]Visualizations were generated using the scikit-learn implementation: https://scikit-learn.org/stable/modules/generated/sklearn.manifold.TSNE.html
[5]https://msnews.github.io/

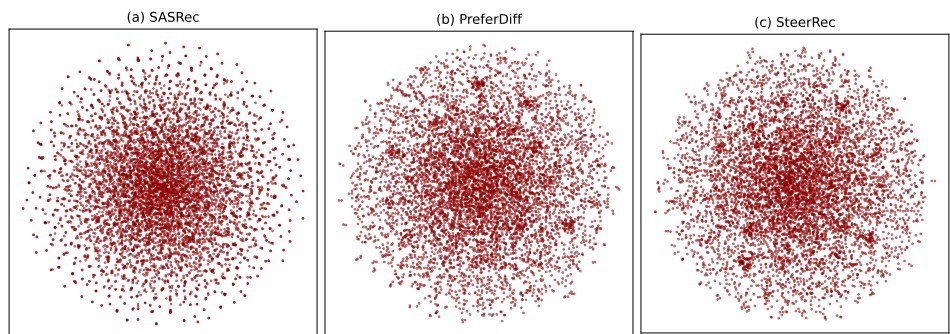

Figure 10: t-SNE visualization of the learned item embedding spaces on the Sports dataset.

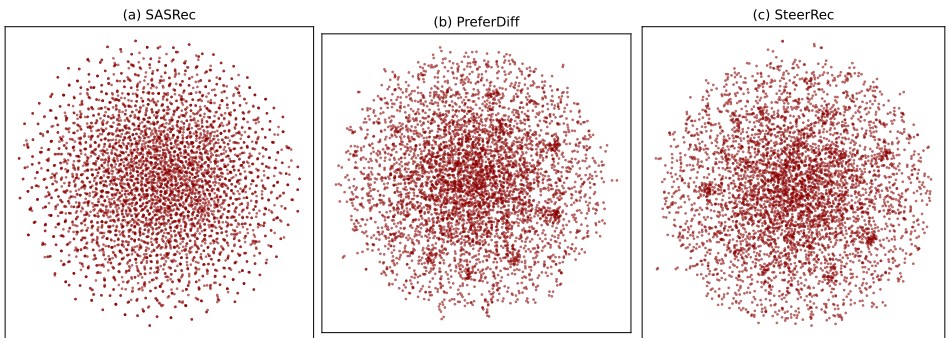

Figure 11: t-SNE visualization of the learned item embedding spaces on the Beauty dataset.

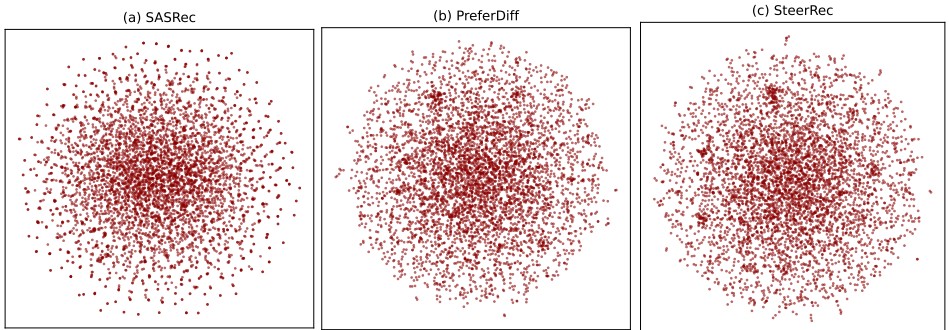

Figure 12: t-SNE visualization of the learned item embedding spaces on the Toys dataset.

## F.1 DATASET DETAILS AND EXPERIMENTAL SETUP

**Dataset Characteristics.** MIND-small is a large-scale benchmark derived from Microsoft News logs. It contains 156,965 training and 73,152 development impression logs. A key feature is the impression-level records: each log includes the clicked article (positive signal) and the simultaneously displayed but unclicked articles, which serve as explicit negative signals. These high-quality negatives make MIND-small an ideal testbed for our PNG mechanism.

**Data Construction.** We process the raw `behaviors.tsv` and `news.tsv` files. User click histories from the "History" field form the positive condition sequence. For each clicked article, the negative condition is constructed from the unclicked items in the same impression. Following the standard protocol, evaluation is performed at the impression level, with the entire impression serving as the candidate set. The official development set is randomly split 50/50 into validation and test subsets.

**Experimental Settings.** All models—PreferDiff, SteerRec with random negatives, and SteerRec with high-quality negatives—share the same SASRec backbone and hyperparameters (Table 6). The embedding dimension is set to 3072, following prior work (Liu et al., 2025). The only differences are in the loss functions and inference-time guidance strategies. Model-specific hyperparameters ($\lambda$ for PreferDiff; $\mu, m$ for SteerRec) are tuned on the validation set.

Table 6: Shared hyperparameter settings for all models on the MIND-small dataset.

| Hyperparameter | Value |
|---|---|
| Backbone Model | SASRec |
| Embedding Dimension ($d$) | 3072 |
| Max Sequence Length | 10 |
| Transformer Heads | 2 |
| Transformer Layers | 1 |
| Optimizer | AdamW |
| Learning Rate | $1 \cdot 10^{-4}$ |
| Batch Size | 256 |
| Guidance Scale ($w$) | 4 |
| Validation Metric | Recall@5 |

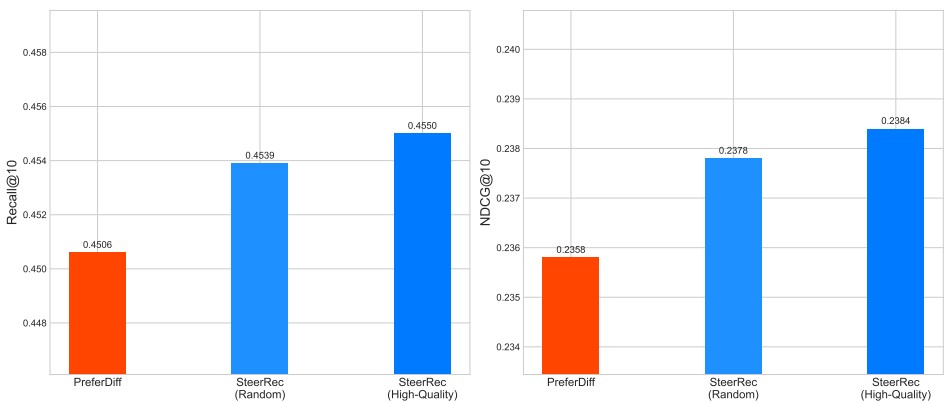

Figure 13: Performance comparison of PreferDiff and SteerRec (with random vs. high-quality negative guidance) on the MIND-small test set.

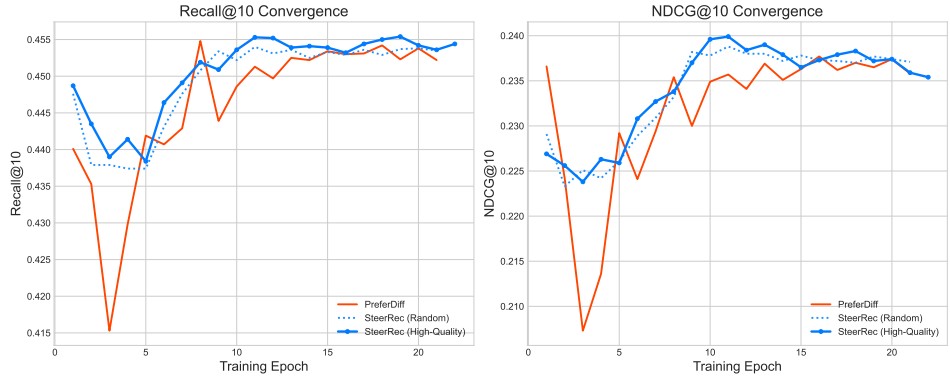

Figure 14: Convergence curves of PreferDiff and SteerRec (with random vs. high-quality negative guidance) on the MIND-small test set.

### F.2 PERFORMANCE ANALYSIS ON MIND-SMALL

To validate our framework on the MIND-small dataset, we analyze PreferDiff against two Steer-Rec variants: one guided by random negatives and one guided by high-quality negatives. This comparison is designed to highlight two core advantages: the fundamental superiority of our PNG mechanism, and the significant performance gains from high-quality, steerable guidance.

**Superiority of the PNG Mechanism.** The first key finding, illustrated in Figure 13, is that Steer-Rec guided by simple random negatives already outperforms PreferDiff. This result validates the core advantage of our PNG mechanism: it directly applies repulsive forces at inference time, overcoming a key limitation of prior work where negative signals are only used indirectly during training.

**Steerability and Stability with High-Quality Guidance.** The second finding highlights the framework's steerable nature. Performance is further amplified when the PNG mechanism is supplied with high-quality, impression-level negatives, achieving the best results. This demonstrates that SteerRec's efficacy scales with the quality of the guidance signal. Moreover, this high-quality guidance also engenders a markedly more stable training process. As shown in Figure 14, this variant avoids the severe initial instability exhibited by other models and converges smoothly, underscoring the dual benefits of providing consistent and meaningful repulsive signals.

## G DISCUSSIONS

### G.1 CORE MOTIVATION: SOLVING TWO FUNDAMENTAL PROBLEMS

Our work is inspired by the success of negative guidance in text-to-image (T2I) models (e.g., Stable Diffusion), which can effectively remove unwanted concepts. To our knowledge, our work is the first to attempt to bring this powerful concept of steerable negative guidance to diffusion-based sequential recommendation. In doing so, we had to solve two distinct problems that prior work did not address:

- **Problem 1: The Training-Inference Inconsistency.** Prior work (like PreferDiff) has a fundamental misalignment: they use negatives in the training loss (BPR) but still rely on a generic null condition (CFG) at inference. Our **Positive-Negative Guidance (PNG)** mechanism solves this by replacing the null condition with a user-aware negative condition $\mathbf{c}^-$, further unlocking the potential of conditional diffusion.

- **Problem 2: The "Semantic Challenge".** Simply applying negative information is suboptimal. As analyzed in our introduction, the T2I paradigm does not transfer directly.
  - *The T2I Paradigm:* Stable Diffusion is trained via the standard CFG paradigm (randomly masking the positive prompt with a null token $\emptyset$). At inference, one can simply swap $\emptyset$ with a negative prompt (e.g., "beard") to achieve negative guidance. This works immediately because T2I relies on **fixed, pre-trained encoders (like CLIP)**, where prompts like "beard" possess stable, universal semantic embeddings that contrast meaningfully with "man".
  - *The RecSys Reality:* Recommendation, conversely, operates in a **dynamic, learned embedding space** where positive conditions, negative conditions, and target items are all drawn from the same evolving item set. This makes negative guidance inherently unstable: simply providing a negative condition $\mathbf{c}^-$ at inference (replacing $\emptyset$) works poorly because the model has not been trained to interpret this signal as a repulsive force. Unlike T2I, where CLIP provides semantic grounding, RecSys models do not inherently know that $\mathbf{c}^-$ means "avoid".
  - *Our Solution:* Our **Guidance Alignment Loss (GAL)** is the explicit training objective designed to solve this semantic challenge. It explicitly builds the geometric separability that CLIP provides "for free" in T2I, teaching the model to treat $\mathbf{c}^-$ as a repulsive signal.

In essence, PNG solves the inconsistency, and GAL solves the semantics. The combination (PNG + GAL) is the complete framework that unlocks the potential of steerable diffusion in recommendation.

## G.2 TRAINING OBJECTIVE AND BASELINE COMPARISON

To better clarify our technical contribution and highlight the novelty of SteerRec, we have prepared a new diagram (Figure 15) that explicitly visualizes our training objective and contrasts it with key diffusion-based recommenders.

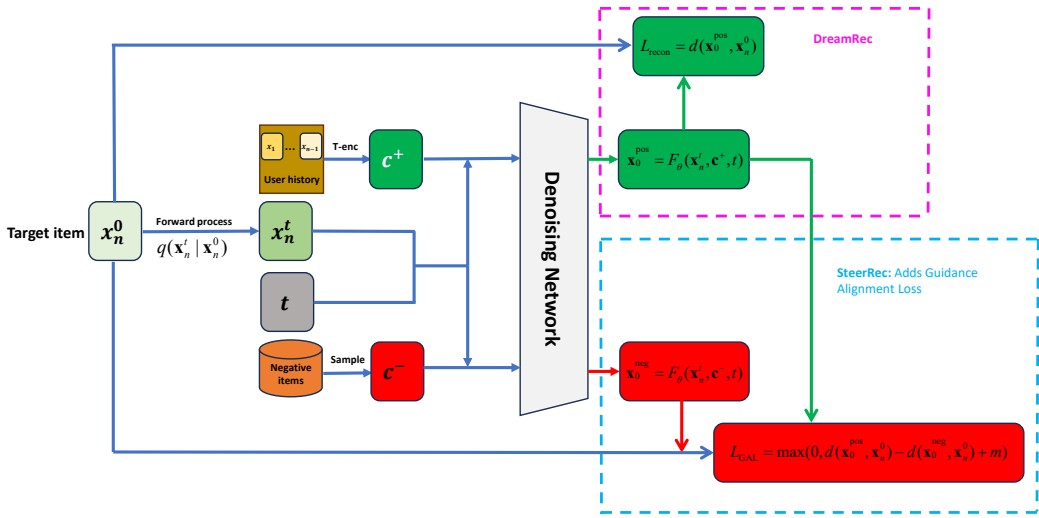

Figure 15: Illustration of the training objective of SteerRec. While DreamRec relies solely on reconstruction conditioned on user history ($\mathbf{c}^+$), SteerRec introduces a parallel negative condition ($\mathbf{c}^-$) and a Guidance Alignment Loss ($L_{\text{GAL}}$). This explicitly trains the denoising network to distinguish between positive and negative guidance signals.

We follow the conditional generation paradigm established by existing works, such as DreamRec (Yang et al., 2023) and PreferDiff (Liu et al., 2025). In this framework, the user history is encoded into a single context vector $\mathbf{c}^+$, and the denoising network $F_\theta$ (typically a simple MLP) is explicitly conditioned on it to predict the target item $\mathbf{x}_n^0$ from the noisy state $\mathbf{x}_n^t$: $F_\theta(\mathbf{x}_n^t, \mathbf{c}^+, t)$. The structural differences lie in how negative information is incorporated:

- **DreamRec (No Negative):** This baseline relies exclusively on a Reconstruction Loss ($L_{\text{recon}}$) based on the positive condition $\mathbf{c}^+$. It does not utilize negative samples during the diffusion training process.

- **PreferDiff (Negative as Target):** This method enhances DreamRec by introducing a ranking loss ($L_{\text{BPR-Diff}}$). Crucially, this approach treats the negative sample as a **Noisy Target** ($\mathbf{x}_t^{neg}$), while the condition remains the user history ($\mathbf{c}^+$).
    - *Mechanism:* $\hat{\mathbf{x}}_0^{neg} = F_\theta(\mathbf{x}_t^{neg}, \mathbf{c}^+, t)$.
    - *Role:* Here, the negative information serves merely as a **reconstruction target** to calculate a BPR ranking loss (comparing the reconstruction error of $\mathbf{x}_t^{pos}$ vs. $\mathbf{x}_t^{neg}$). The model never learns to generate based on a negative condition. Consequently, at inference, it cannot utilize negative signals for guidance and must revert to standard CFG.

- **SteerRec (Negative as Condition):** Our training objective integrates the Guidance Alignment Loss ($L_{\text{GAL}}$). In stark contrast, SteerRec inputs the negative information as a **Guidance Condition** ($\mathbf{c}^-$), acting upon the positive noisy target ($\mathbf{x}_t^{pos}$).
    - *Mechanism:* $\hat{\mathbf{x}}_0^{neg} = F_\theta(\mathbf{x}_t^{pos}, \mathbf{c}^-, t)$.
    - *Role:* Here, the negative information serves as an **active guidance signal**. We explicitly train the network to output a "repulsive" prediction when conditioned on $\mathbf{c}^-$. This structural shift is what enables our PNG mechanism to actively steer generation at inference time, a capability PreferDiff fundamentally lacks.

As illustrated in Figure 15, this alignment allows the network to interpret $\mathbf{c}^-$ as a repulsive signal, directly serving our Positive-Negative Guidance (PNG) mechanism at inference.

### G.3 GENERALIZABILITY TO DATASETS WITH LONGER SEQUENCES

A key question is how well SteerRec generalizes to datasets characterized by longer user interaction histories, as the Amazon datasets used in the main experiments have relatively short average sequence lengths ($< 10$). To address this, we conducted additional experiments on the widely-used MovieLens-1M (ML-1M) benchmark. Following the protocol of recent diffusion-based recommenders (Liu et al., 2025), we varied the maximum sequence length from 10 to 50. The results are presented in Table 7.

Table 7: Recommendation Performance (Recall@5 / NDCG@5) with varied length of user history on ML-1M. The best performance is in **bold**.

| Model | Length=10 | Length=20 | Length=30 | Length=40 | Length=50 |
|---|---|---|---|---|---|
| SASRec | 0.0201 / 0.0137 | 0.0242 / 0.0131 | 0.0306 / 0.0179 | 0.0217 / 0.0138 | 0.0205 / 0.0134 |
| BERT4Rec | 0.0215 / 0.0152 | 0.0265 / 0.0146 | 0.0331 / 0.0200 | 0.0248 / 0.0154 | 0.0198 / 0.0119 |
| TIGIR | 0.0451 / 0.0298 | 0.0430 / 0.0270 | 0.0430 / 0.0289 | 0.0364 / 0.0238 | 0.0430 / 0.0276 |
| DreamRec | 0.0464 / 0.0314 | 0.0480 / 0.0349 | 0.0514 / 0.0394 | 0.0497 / 0.0350 | 0.0447 / 0.0377 |
| PreferDiff | 0.0629 / 0.0439 | 0.0513 / 0.0365 | 0.0546 / 0.0408 | **0.0596** / 0.0420 | 0.0546 / 0.0399 |
| **SteerRec** | **0.0728 / 0.0531** | **0.0679 / 0.0525** | **0.0596 / 0.0466** | 0.0579 / **0.0423** | **0.0646 / 0.0469** |

**Results.** As shown in Table 7, SteerRec consistently outperforms other baselines across different lengths of user historical interactions. This confirms the robustness and generalizability of our framework beyond short-sequence scenarios.

### G.4 ANALYSIS OF INFERENCE EFFICIENCY AND TIMESTEPS

We analyze the trade-off between diffusion timesteps ($T$) and model performance, as well as the trade-off between inference steps ($S$) and efficiency.

**1. Training Timesteps ($T$) vs. Performance.** A large number of training timesteps is necessary for current diffusion-based recommenders. This aligns with findings in related work (Liu et al., 2025), which suggest that high-dimensional embeddings ($D = 3072$) require a fine-grained denoising process (i.e., large $T$) to learn the complex distribution effectively. To validate this, we conducted an ablation study on $T$ while fixing the number of inference steps at $S = 10$.

Table 8: Effect of different *training* timesteps $T$ on performance (Recall@5 / NDCG@5), with inference steps fixed at $S = 10$.

| Training Steps ($T$) | Sports | Beauty | Toys |
|---|---|---|---|
| 100 | 0.0112 / 0.0072 | 0.0148 / 0.0094 | 0.0216 / 0.0164 |
| 200 | 0.0121 / 0.0076 | 0.0215 / 0.0153 | 0.0314 / 0.0236 |
| 400 | 0.0146 / 0.0101 | 0.0331 / 0.0246 | 0.0443 / 0.0320 |
| 600 | 0.0202 / 0.0153 | 0.0376 / 0.0284 | 0.0458 / 0.0339 |
| 800 | 0.0205 / 0.0153 | 0.0380 / 0.0286 | 0.0473 / 0.0370 |
| 1000 | 0.0208 / 0.0167 | 0.0429 / 0.0325 | 0.0465 / 0.0363 |
| 1200 | 0.0199 / 0.0148 | 0.0443 / 0.0334 | 0.0461 / 0.0359 |
| 2000 | 0.0205 / 0.0145 | 0.0416 / 0.0322 | 0.0453 / 0.0337 |
| 4000 | 0.0208 / 0.0155 | 0.0420 / 0.0330 | 0.0463 / 0.0367 |

**Results.** We observe that recommendation performance drops significantly when the training timesteps $T$ are too small (e.g., $T < 400$). The model generally achieves optimal performance

in the range of $T = 800$ to $1200$. This empirically confirms our hypothesis that a fine-grained discretization (i.e., a sufficiently large $T$) is essential for learning accurate denoising trajectories in the high-dimensional item embedding space ($D = 3072$).

**2. Inference Steps ($S$) vs. Efficiency.** While training requires large $T$, inference can be highly efficient. By using the DDIM sampler, we can skip steps during generation. For instance, on the Sports dataset ($T = 1000$), setting the skip interval to 100 results in only 10 actual denoising steps ($S = 10$), which takes approximately 2 seconds per batch. Table 9 demonstrates the trade-off between inference time and performance. SteerRec achieves commendable results with as few as 10 to 20 steps.

Table 9: Effect of different *denoising* (inference) steps $S$ on performance (Recall@5 / NDCG@5). Transposing the table allows for a clearer comparison of the trade-off between inference time and recommendation quality.

| Inference Steps ($S$) | Time Cost | Performance (Recall@5 / NDCG@5) | | |
|:---:|:---:|:---:|:---:|:---:|
| | | **Sports** | **Beauty** | **Toys** |
| 1 | <1s | 0.0186 / 0.0145 | 0.0407 / 0.0310 | 0.0422 / 0.0339 |
| 2 | <1s | 0.0193 / 0.0143 | 0.0407 / 0.0304 | 0.0459 / 0.0358 |
| 5 | 1s | 0.0199 / 0.0156 | 0.0416 / 0.0320 | 0.0456 / 0.0353 |
| 10 | 2s | 0.0208 / 0.0167 | 0.0429 / 0.0325 | 0.0461 / 0.0363 |
| 20 | 3s | 0.0210 / 0.0167 | 0.0433 / 0.0331 | 0.0470 / 0.0370 |
| 50 | 12s | 0.0211 / 0.0164 | 0.0443 / 0.0334 | 0.0473 / 0.0370 |
| 100 | 23s | 0.0211 / 0.0163 | 0.0442 / 0.0330 | 0.0469 / 0.0367 |
| 500 | 57s | 0.0213 / 0.0163 | 0.0445 / 0.0334 | 0.0473 / 0.0370 |
| 1000 | 120s | 0.0213 / 0.0162 | 0.0443 / 0.0334 | 0.0473 / 0.0370 |

**Results.** As demonstrated in the table, SteerRec achieves competitive performance with as few as 10 to 20 inference steps, requiring only roughly 2–3 seconds per batch. While increasing the steps to 1000 yields slight improvements in some metrics, the computational cost increases linearly (from 3s to 120s), which is disproportionate to the marginal performance gain. Therefore, setting the inference steps $S \in [10, 20]$ provides an optimal balance between recommendation accuracy and real-world efficiency.

### G.5 TIME COMPLEXITY AND RUNTIME ANALYSIS

To demonstrate the practicality of SteerRec, we analyze its theoretical time complexity and compare its actual runtime against key baselines.

**Theoretical Time Complexity Analysis.** We analyze the time complexity per training/inference batch. Let $B$ denote the batch size, $L$ the sequence length, $D$ the hidden embedding dimension, $N$ the total number of items, and $S$ the number of inference steps.

- **Training:** The training complexity consists of two main components:
  - **Encoder (Transformer):** The complexity for the self-attention mechanism is $O(B \cdot L^2 \cdot D)$.
  - **Denoising (MLP):** Our denoising network requires two forward passes (one for $\mathbf{c}^+$ and one for $\mathbf{c}^-$) to calculate $L_{\mathrm{GAL}}$. Since the network consists of linear layers, the complexity is proportional to $O(B \cdot D^2)$.
  - **Comparison:** This is asymptotically identical to PreferDiff, which also requires a Transformer pass and two denoising passes (for positive and negative items) to compute its ranking loss.
- **Inference:** The inference complexity consists of the following components:
  - **Encoder:** Encoding the user history takes $O(B \cdot L^2 \cdot D)$.
  - **Guidance Mechanism:** The PNG mechanism requires two forward passes per DDIM step (Total: $2 \cdot S$ passes). The complexity is $2 \cdot S \cdot O(B \cdot D^2)$. This is exactly the same

as the standard CFG mechanism used by baselines, which also requires two passes ($\mathbf{c}^+$ and $\emptyset$).

- **Full Ranking:** Calculating scores for all items takes $O(B \cdot N \cdot D)$.
- **Negative Sampling:** The random negative sampling is a simple lookup operation with negligible cost compared to the matrix operations above.
- **Conclusion:** Since all major complexity terms ($O(S \cdot B \cdot D^2)$ and $O(B \cdot N \cdot D)$) are present in both methods, SteerRec introduces no additional asymptotic complexity compared to standard CFG-based diffusion models.

**Actual Runtime Comparison.** Furthermore, we also make comparisons of training time and inference time between SteerRec and other baselines.

Table 10: Comparison of Training Time and Inference Times.

| Dataset | Model | Training Time (s/epoch) / (s/total) | Inference Time (s/epoch) |
|---------|-------|-------------------------------------|--------------------------|
| Sports | SASRec | 2.67 / 31 | 0.47 |
| | BERT4Rec | 7.87 / 79 | 0.65 |
| | TIGER | 11.42 / 1069 | 24.14 |
| | DreamRec | 16.32 / 811 | 356.43 |
| | PreferDiff | 20.78 / 588 | 2.11 |
| | SteerRec | 21.15 / 416 | 2.23 |
| Beauty | SASRec | 1.05 / 36 | 0.37 |
| | BERT4Rec | 3.66 / 80 | 0.40 |
| | TIGER | 5.41 / 1058 | 10.19 |
| | DreamRec | 10.36 / 535 | 288.32 |
| | PreferDiff | 13.00 / 470 | 1.62 |
| | SteerRec | 13.22 / 364 | 1.81 |
| Toys | SASRec | 0.80 / 56 | 0.22 |
| | BERT4Rec | 3.11 / 93 | 0.23 |
| | TIGER | 3.76 / 765 | 4.21 |
| | DreamRec | 10.13 / 512 | 302.75 |
| | PreferDiff | 12.07 / 437 | 1.29 |
| | SteerRec | 12.21 / 412 | 1.36 |

**Results.** We can observe that while SteerRec maintains comparable inference and per-epoch training times to PreferDiff, it demonstrates a significant advantage in Total Training Time. This is because our GAL can construct user preferences more effectively, leading to much faster convergence.

### G.6 IMPACT OF NEGATIVE SAMPLING STRATEGIES

In the early stages of this research, we explicitly explored **Retrieval-based Hard Negative Mining (HNM)** at inference. Specifically, we employed a strategy of ranking all items based on similarity to the user context (i.e., the encoded user history vector $\mathbf{c}^+$). We selected $N_{neg} = 20$ negative samples from different similarity intervals: the strict top ranks (Top 20), the top percentile (Top 1%-3%), and a mid-high percentile (Top 10%-15%).

Table 11: Performance and Efficiency of Different Negative Sampling Strategies (Sports Dataset).

| Negative Strategy | R@5 | N@5 | R@10 | N@10 | Time (s) |
|-------------------|------|------|------|------|----------|
| Random | 0.0208 | 0.0167 | 0.0275 | 0.0189 | 2.2s |
| HNM (Top 1%-3%) | 0.0211 | 0.0169 | 0.0280 | 0.0191 | 3.6s |
| HNM (Top 10%-15%) | 0.0209 | 0.0166 | 0.0276 | 0.0189 | 3.6s |
| HNM (Top 20) | 0.0185 | 0.0152 | 0.0262 | 0.0177 | 3.6s |

We identified two critical issues:

- **A. The "False Negative" Risk:** As shown in Table 11, strict HNM (Top 20) causes a performance drop. This is because high-similarity negatives likely include the Target Item (false negative). Treating the ground truth as a negative condition "poisons" the repulsive guidance, actively steering the model away from the correct target.

- **B. Scalability Bottleneck:** Random sampling is $O(1)$, whereas HNM requires expensive full-corpus ranking ($O(N)$).
    - *Training Cost:* Unlike inference where the HNM cost is amortized over multiple denoising steps, training involves only *one* network step per batch. Thus, adding Global HNM would roughly double the training time per epoch, making it computationally prohibitive on large datasets.

**Conclusion:** Random Sampling currently offers the best balance of robustness and efficiency, though developing efficient HNM strategies remains a promising avenue for future work.

