# OpenReview forum: "Sculpting User Preferences for Recommendation with Positive-Negative Diffusion Guidance"
_ICLR.cc/2026/Conference — Submitted to ICLR 2026_

### Official Review · Reviewer_kUAa · 2025-10-31

**Soundness:** 3
**Presentation:** 3
**Contribution:** 2
**Rating:** 4
**Confidence:** 5

**Summary:**

The paper introduces SteerRec, a novel framework for sequential recommendation that leverages diffusion models (DMs) for more accurate and personalized predictions. Traditional diffusion-based recommenders struggle with incorporating negative feedback during inference, as the existing classifier-free guidance (CFG) mechanism relies on a global, user-agnostic prior, which limits control over the generative process. SteerRec addresses this limitation by introducing Positive-Negative Guidance (PNG), which uses user-aware negative feedback to explicitly steer the generation process away from undesired items. This is complemented by the Guidance Alignment Triplet Loss (GAL), a novel training objective that ensures the model generates predictions that are both closer to desired items and farther from disliked ones. Through extensive experiments on public benchmarks, SteerRec outperforms existing methods, providing a more precise and controllable recommendation system.

**Strengths:**

1. SteerRec's key contribution is the Positive-Negative Guidance (PNG) mechanism based on the well-known classifier-free guidance, which directly integrates user-aware negative feedback into the inference process of diffusion model-based recommenders, providing enhanced control over the generation process. To further optimize the proposed of PNG, the authors introduce the Guidance Alignment Triplet Loss which ensures that the negative guidance is both meaningful and effective during training. This alignment makes the model's predictions more reliable and better aligned with user preferences. The combination of both modules is both logical and effective.

2. SteerRec converges faster than its counterparts like PreferDiff, demonstrating that its direct and efficient learning signal enhances the model's ability to learn user preferences quickly.

**Weaknesses:**

1. **Paper Writing:** A major issue in the paper's writing is the lack of a detailed comparison between SteerRec and previous methods (e.g., DreamRec [1], PreferDiff [2]). This omission assumes that readers are already familiar with diffusion-based recommenders, which increases the reading burden for those who may not have such background knowledge. Further

2. **Brute Force Sampling:** I think the author's motivation is both reasonable and compelling. Replacing CFG with recommendation-tailored guidance is essential. However, in practice, the method of constructing the negative condition during inference by randomly sampling a set of items from the global item corpus seems somewhat crude. This approach overlooks the unique preferences encoded in the user's historical representation, potentially leading to less precise guidance. I think more tailored approaches for selecting negative samples that are more in line with the principles of diffusion-based recommenders.   I also noticed in Appendix F that the authors' experiments show SteerRec performing well when high-quality negative samples from the MIND dataset are used. **However, obtaining high-quality negative samples is challenging in most real-world scenarios.** Addressing this issue should be a key focus of the paper, as both PNG and GAL are quite intuitive ideas. A potential solution could involve methods to generate or identify high-quality negative samples in settings where they are not readily available.

3. **Missing Some Hyperparameter Experiments** I find some key experiments related to diffusion, such as the impact of the diffusion steps and DDIM steps, were not presented. If SteerRec requires fewer diffusion or DDIM steps compared to PreferDiff, it would further highlight the efficiency brought by the introduction of negative guidance. Showing these results would provide stronger evidence for the practical benefits of SteerRec's approach.

4. **Need More Novelty on Negative Guidance** The use of relevant negative signals aggregated into a centroid is shown to be effective in PreferDiff, but this approach lacks some novelty in SteerRec. A potential improvement could be to explore why other users' representations are not used as negative samples. Incorporating negative samples from different users might provide richer and more diverse guidance.


[1] Yang, Zhengyi, et al. "Generate what you prefer: Reshaping sequential recommendation via guided diffusion." Advances in Neural Information Processing Systems 36 (2023): 24247-24261.

[2] Liu, Shuo, et al. "Preference Diffusion for Recommendation." The Thirteenth International Conference on Learning Representations.

**Questions:**

See weaknesses.

---

> ### Author Response · Authors · 2025-11-20
> **Response to Reviewer kUAa - Part (1/4)**
>
> Thank you for your insightful review and for your excellent summary of our work. You have raised several critical and fair points. We appreciate this constructive feedback, which has prompted us to provide much clearer justifications. Below are our detailed responses, which we believe fully address your concerns. We would be delighted to engage further with you if you have any additional feedback.
>
> > **Comment 1: Paper Writing** — “A major issue in the paper's writing is the lack of a detailed comparison between SteerRec and previous methods (e.g., DreamRec, PreferDiff). This omission assumes that readers are already familiar with diffusion-based recommenders... increases the reading burden.”
>
> Thank you for this important feedback regarding the paper's writing and its accessibility to readers who may not be experts in diffusion-based recommenders. We appreciate this concern and would like to assure you that this background and comparison is indeed included in the paper, though we agree it could be made more prominent. Specifically, we placed:
> * Detailed technical background on DMs and CFG in **Section 2 (Preliminary)**.
> * A direct conceptual comparison in the **Introduction** (in the second paragraph).
> * A thorough literature review differentiating our work from others in **Appendix B (Related Work)**.
>
> To centralize this comparison and directly address your concern, we have provided a more detailed elaboration of the differences between SteerRec and existing works (DreamRec, PreferDiff) in **Appendix G.2**. We will also further revise the Related Work section in the final version to explicitly reflect these contrasts, ensuring the distinction is clear for all readers.

---

> ### Author Response · Authors · 2025-11-20
> **Response to Reviewer kUAa - Part (2/4)**
>
> > **Comment 2: Brute Force Sampling** — “...the method of constructing the negative condition during inference by randomly sampling a set of items... seems somewhat crude. This approach overlooks the unique preferences encoded in the user's historical representation... potentially leading to less precise guidance.”
>
> Thank you for this excellent and insightful question. You have correctly identified a key aspect of our experimental design. Our choice of simple sampling was a deliberate one, which we believe highlights the core strengths of our framework. Here is our reasoning:
>
> **1. Core Motivation: Unlocking Steerable Negative Guidance**
> Our work is inspired by the success of negative guidance in text-to-image (T2I) models (e.g., Stable Diffusion), which can effectively remove unwanted concepts (e.g., "generate a man *without a beard*"). To our knowledge, we are the first work to explore how to bring this powerful negative guidance to recommendation to further unlock the potential of diffusion models.
> Our primary design focus was not to propose a complex negative sampling strategy, but to design the **PNG inference mechanism** and **GAL training objective** themselves. Without PNG and GAL, there is no mechanism for diffusion models to utilize *any* negative signal (whether random or hard) during generation.
> While the concepts of PNG and GAL may appear intuitive at first glance, their design is far from trivial and is underpinned by a profound insight into the fundamental differences between text-to-image generation and recommendation systems. A detailed analysis of this motivation is provided in **Appendix G.1**.
>
> **2. Robustness and Standard Practice:**
> By showing that simple sampling works, we prove that the performance gains stem from our controllable negative guidance framework rather than a complex sampling trick. Previous seminal works, such as SASRec [Kang et al., 2018], CL4SRec [Xie et al., 2022], and PreferDiff [Liu et al., 2025], have demonstrated that simple negative sampling (e.g., random or in-batch sampling) is both effective and standard practice in recommendation systems due to data sparsity (most items are true negatives).
>
> **3. The Pitfalls of Hard Negative Mining (HNM):**
> In the early stages of this research, we explicitly explored Retrieval-based Hard Negative Mining (HNM). Specifically, we employed a strategy of ranking all items based on similarity to the user context (i.e., the encoded user history vector $\mathbf{c}^+$). We selected $N_{neg}=20$ negative samples from different similarity intervals: the strict top ranks (Top 20), the top percentile (Top 1%-3%), and a mid-high percentile (Top 10%-15%).
>
> **Table 1: Performance and Efficiency of Different Negative Sampling Strategies (Sports Dataset)**
>
> | Negative Strategy | R@5 | N@5 | R@10 | N@10 | Time (s) |
> | :--- | :--- | :--- | :--- | :--- | :--- |
> | **Random** | 0.0208 | 0.0167 | 0.0275 | 0.0189 | 2.2s |
> | **HNM (Top 1%-3%)** | 0.0211 | 0.0169 | 0.0280 | 0.0191 | 3.6s |
> | **HNM (Top 10%-15%)**| 0.0209 | 0.0166 | 0.0276 | 0.0189 | 3.6s |
> | **HNM (Top 20)** | 0.0185 | 0.0152 | 0.0262 | 0.0177 | 3.6s |
>
> We identified two critical issues:
> * **A. The "False Negative" Risk:** As shown in Table 1, strict HNM (Top 20) causes a performance drop. This is because high-similarity negatives likely include the Target Item (false negative). Treating the ground truth as a negative condition "poisons" the repulsive guidance, actively steering the model away from the correct target.
> * **B. Scalability Bottleneck:** Random sampling is $O(1)$, whereas HNM requires expensive full-corpus ranking ($O(N)$).
>     * **Training Cost:** Unlike inference where the HNM cost is amortized over multiple denoising steps, training involves only *one* network step per batch. Thus, adding Global HNM would roughly double the training time per epoch, making it computationally prohibitive on large datasets.
>
> **Conclusion:** Random Sampling currently offers the best balance of robustness and efficiency, though developing efficient HNM strategies remains a promising avenue for future work.
>
> [1] Kang, Wang-Cheng, et al. "Self-attentive sequential recommendation." Proceedings of the 18th IEEE International Conference on Data Mining. 2018.
>
> [2] Xie, Xu, et al. "Contrastive learning for sequential recommendation." Proceedings of the 38th IEEE International Conference on Data Engineering. 2022.
>
> [3] Liu, Shuo, et al. "Preference Diffusion for Recommendation." The Thirteenth International Conference on Learning Representations. 2025.

---

> ### Author Response · Authors · 2025-11-20
> **Response to Reviewer kUAa - Part (3/4)**
>
> > **Comment 3: Missing Some Hyperparameter Experiments** — “I find some key experiments related to diffusion, such as the impact of the diffusion steps and DDIM steps, were not presented... Showing these results would provide stronger evidence for the practical benefits of SteerRec's approach.”
>
> Thank you for this excellent suggestion. We have conducted additional experiments on the impact of timesteps and denoising steps. We will add these results to Appendix G.4 in the revised paper.
>
> **1. Training Timesteps ($T$) and Performance Trade-off**
> A large number of training timesteps is necessary for current diffusion-based recommenders. This aligns with findings in related work (e.g., PreferDiff), which suggest that high-dimensional embeddings ($D=3072$) require a fine-grained denoising process (i.e., large $T$) to learn the complex distribution effectively. To validate this, we conducted an ablation study on $T$ while fixing the number of inference steps at $S=10$.
>
> **Table 2: Effect of different *training* timesteps $T$ (Recall@5 / NDCG@5)**
>
> | $T$ | Sports | Beauty | Toys |
> | :--- | :--- | :--- | :--- |
> | 100 | 0.0112 / 0.0072 | 0.0148 / 0.0094 | 0.0216 / 0.0164 |
> | 200 | 0.0121 / 0.0076 | 0.0215 / 0.0153 | 0.0314 / 0.0236 |
> | 400 | 0.0146 / 0.0101 | 0.0331 / 0.0246 | 0.0443 / 0.0320 |
> | 600 | 0.0202 / 0.0153 | 0.0376 / 0.0284 | 0.0458 / 0.0339 |
> | 800 | 0.0205 / 0.0153 | 0.0380 / 0.0286 | 0.0473 / 0.0370 |
> | 1000 | 0.0208 / 0.0167 | 0.0429 / 0.0325 | 0.0465 / 0.0363 |
> | 1200 | 0.0199 / 0.0148 | 0.0443 / 0.0334 | 0.0461 / 0.0359 |
> | 2000 | 0.0205 / 0.0145 | 0.0416 / 0.0322 | 0.0453 / 0.0337 |
> | 4000 | 0.0208 / 0.0155 | 0.0420 / 0.0330 | 0.0463 / 0.0367 |
>
> **Results:** As shown in Table 2, training with a sufficiently large $T$ (e.g., 1000) is necessary to ensure fine-grained learning of the high-dimensional embedding space.
>
> **2. Inference Steps ($S$) and Efficiency Trade-off**
> To validate efficiency, we fixed training $T=1000$ and varied the number of *denoising (inference) steps* $S$ using the DDIM sampler.
>
> **Table 3: Effect of different *denoising* (inference) steps $S$ (Recall@5 / NDCG@5)**
>
> | SteerRec (Recall@5/NDCG@5) | 1 (<1s) | 2 (<1s) | 5 (1s) | 10 (2s) | 20 (3s) | 50 (12s) | 100 (23s) | 500 (57s) | 1000 (120s) |
> | :--- | :--- | :--- | :--- | :--- | :--- | :--- | :--- | :--- | :--- |
> | **Sports** | 0.0186/0.0145 | 0.0193/0.0143 | 0.0199/0.0156 | 0.0208/0.0167 | 0.0210/0.0167 | 0.0211/0.0164 | 0.0211/0.0163 | 0.0213/0.0163 | 0.0213/0.0162 |
> | **Beauty** | 0.0407/0.0310 | 0.0407/0.0304 | 0.0416/0.0320 | 0.0429/0.0325 | 0.0433/0.0331 | 0.0443/0.0334 | 0.0442/0.0330 | 0.0445/0.0334 | 0.0443/0.0334 |
> | **Toys**  | 0.0422/0.0339 | 0.0459/0.0358 | 0.0456/0.0353 | 0.0461/0.0363 | 0.0470/0.0370 | 0.0473/0.0370 | 0.0469/0.0367 | 0.0473/0.0370 | 0.0473/0.0370 |
>
> **Conclusion:** SteerRec achieves optimal performance with just 10-20 steps. Crucially, its superiority over PreferDiff stems from the Guidance Alignment Loss (GAL), not standard diffusion settings (DDIM/Timesteps). Unlike PreferDiff's indirect BPR loss, GAL enforces a direct geometric constraint, enabling significantly faster convergence and better preference modeling.

---

> ### Author Response · Authors · 2025-11-20
> **Response to Reviewer kUAa - Part (4/4)**
>
> > **Comment 4: Need More Novelty on Negative Guidance** — “The use of relevant negative signals aggregated into a centroid is shown to be effective in PreferDiff, but this approach lacks some novelty in SteerRec. ... Incorporating negative samples from different users might provide richer and more diverse guidance.”
>
> Thank you for this insightful suggestion. We acknowledge that utilizing a centroid of negatives is a shared technique. However, we believe there is a fundamental misunderstanding regarding **how this centroid is utilized** within the diffusion architecture. The novelty lies not in the aggregation itself, but in its **structural role** within the denoising network:
>
> * **PreferDiff (Negative as Target):** PreferDiff inputs the negative centroid into the network as the **Noisy Target ($\mathbf{x}_t^{neg}$)**, while keeping the condition as the user history ($\mathbf{c}^+$).
>     * *Mechanism:* $\hat{\mathbf{x}} _ 0^{neg} = F _ \theta(\mathbf{x} _ t^{neg}, \mathbf{c}^+, t)$.
>     * *Role:* Here, the negative centroid serves merely as a **reconstruction target** to calculate a BPR ranking loss. The model never learns to generate *based on* a negative condition. Consequently, at inference, it cannot utilize negative signals for guidance and must revert to standard CFG.
>
> * **SteerRec (Negative as Condition):** In contrast, SteerRec inputs the negative centroid into the network as the **Condition ($\mathbf{c}^-$)**, acting upon the positive noisy target ($\mathbf{x}_t^{pos}$).
>     * *Mechanism:* $\hat{\mathbf{x}} _ 0^{neg} = F _ \theta(\mathbf{x} _ t^{pos}, \mathbf{c}^-, t)$.
>     * *Role:* Here, the negative centroid serves as an **active guidance signal**. We explicitly train the network to output a "repulsive" prediction when conditioned on $\mathbf{c}^-$. This structural shift is what enables our PNG mechanism to actively steer generation at inference time, a capability PreferDiff fundamentally lacks.
>
> Regarding your suggestion to use "other users' representations" as negative samples: this is a fascinating idea. However, in our current conditional diffusion architecture, the generation target is an Item Embedding, not a User Representation. Therefore, constructing the negative condition using Items is more direct and aligned with the feature space. Nevertheless, incorporating collaborative negative signals from other users is a promising direction for future work.

---

> ### Author Response · Authors · 2025-11-24
> **looking forward to your reply**
>
> Dear Reviewer kUAa,
>
> Thank you again for your constructive comments. We have submitted detailed responses to clarify the comparison with baselines and provided the additional hyperparameter experiments you requested.
>
> We look forward to further discussion with you and would greatly appreciate your feedback on our rebuttal.
>
> Best regards,
>
> The Authors

---

> ### Comment · Reviewer_kUAa · 2025-11-27
>
> Thank you for your valuable efforts in the rebuttal.
>
> 1. I think the paper can be further strengthened by positioning it from the perspectives of PreferDiff (Negative as Target) and SteerRec (Negative as Condition). This framing can more clearly articulate the distinctions from prior studies and highlight the core contributions of the work.
>
> 2. Thank you very much for the comprehensive negative sampling experiments. I find them highly thorough, and they should definitely be included in the paper. The relatively modest performance of HNM also appears to be consistent with expectations.
>
> > Regarding your suggestion to use "other users' representations" as negative samples: this is a fascinating idea. However, in our current conditional diffusion architecture, the generation target is an Item Embedding, not a User Representation.
>
> 3. I agree with some of the authors’ points. However, although the generation target is an item embedding, the user’s sequential features still function as conditions that influence the entire denoising process. Since the goal is to construct a negative condition, it seems more consistent to start from a negative user sequence representation. But using the mean of multiple sampled items is certainly reasonable—as it can be interpreted as a representation of a hypothetical user sequence—but it may not fully capture the notion of a hard negative.
>
> I think one possible alternative is to construct a fake user sequence, e.g. by **masking certain positions** in the sequence and using the resulting representation as a negative condition. This may also serve as a meaningful form of negative conditioning. These are, of course, just some of my thoughts.
>
> My view is that further exploring this aspect is essential, as it represents the core of negative-condition approaches such as SteerRec. Understanding how to construct and manipulate meaningful negative conditions may hold the key to fully leveraging this line of methodology.
>
> Anyway, I sincerely appreciate the authors’ efforts during the rebuttal stage. The overall idea of the paper is also quite good, and I will raise my score accordingly, assuming that the writing can be improved.

---

> > ### Author Response · Authors · 2025-11-28
> >
> > Dear Reviewer kUAa,
> >
> > We are genuinely grateful for your encouraging feedback and your decision to raise the score.
> >
> > We are fully committed to polishing our manuscript, particularly by adopting your insightful framing of "Negative as Target" vs. "Negative as Condition" to clarify our contributions. We also find your suggestion on constructing negative conditions via **masked user sequences** highly illuminating; we will conduct experiments on this promising direction and look forward to further discussion with you.
> >
> > Best regards,
> >
> > The Authors

---

### Official Review · Reviewer_tTUQ · 2025-11-01

**Soundness:** 2
**Presentation:** 3
**Contribution:** 2
**Rating:** 4
**Confidence:** 4

**Summary:**

This paper focuses on the improvement of conditional guidance strategies for diffusion-based recommender systems. Specifically, the original classifier-free guidance simultaneously models both the conditional and unconditional distributions' score functions (log-likelihood gradients), thereby using the weighted difference between the conditional and unconditional score functions as an additional guidance condition during generation. In contrast, this paper replaces the unconditional vector (a trainable embedding) with a weighted embedding derived from in-batch negative samples, thereby introducing negative conditional guidance. Furthermore, this paper introduces a Guidance Alignment Triplet Loss to further regularize negative guidance. The proposed method is relatively simple and effective. However, on the one hand, it completely discards modeling the unconditional distribution, which may affect the model’s cold-start capability. On the other hand, the need for negative sampling during inference could further impact the inference efficiency of diffusion models.

**Strengths:**

1. This  paper is well-organized, with clear tables and figures.
2. The motivation of this paper is well-founded, and the method is simple yet effective.
3. The experimental setup in this paper is well-aligned with prior work and relatively extensive.

**Weaknesses:**

1. **Lack of diverse negative sampling strategies: **The computation of the negative condition in this paper relies on negative sampling; however, only in-batch negative sampling is considered. Exploring more diverse and fine-grained negative condition constructions—such as incorporating hard negatives—could further enrich the content and strengthen the contributions of the paper.
2. **Lack of cold-start analysis:** This paper completely replaces the *none condition* with a *negative condition*. However, the computation of the negative condition relies on negative sampling (e.g., in-batch negative sampling). For cold-start users, whose positive interaction information is relatively scarce, the influence of negative signals becomes more significant, and the use of random negative sampling may adversely affect recommendation performance. Nevertheless, the paper does not include any discussion or analysis regarding the cold-start problem.
3. **Training–inference inconsistency:** The proposed method computes the negative condition during training using in-batch negatives; however, batch information is unavailable during inference, so only random negative sampling can be applied. In this case, if positive items are accidentally sampled as negatives, it may undermine the validity of the negative condition.
4. **Lack of efficiency analysis:** The proposed method requires explicit random negative sampling during inference to compute the negative condition, which could further reduce the model’s recommendation efficiency and even affect its applicability in online settings. However, the paper does not provide any comparative analysis of efficiency.

**Questions:**

Please refer to Weakness.

---

> ### Author Response · Authors · 2025-11-20
> **Response to Reviewer tTUQ - Part (1/4)**
>
> Thank you for your insightful review and for your accurate summary of our work. We truly appreciate your thoughtful feedback and the four essential weaknesses you raised. Below are our detailed responses, which we believe have fully addressed all your questions. We would be delighted to engage further with you if you have any additional feedback.
>
> > **Comment 1: Lack of diverse negative sampling strategies** — "The computation of the negative condition in this paper relies on negative sampling; however, only in-batch negative sampling is considered. Exploring more diverse and fine-grained negative condition constructions—such as incorporating hard negatives—could further enrich the content and strengthen the contributions of the paper."
>
> Thank you for this excellent and insightful question. You have correctly identified a key aspect of our experimental design. Our choice of simple sampling was a deliberate one, which we believe highlights the core strengths of our framework. Here is our reasoning:
>
> **1. Core Motivation: Unlocking Steerable Negative Guidance**
> Our work is inspired by the success of negative guidance in text-to-image (T2I) models (e.g., Stable Diffusion), which can effectively remove unwanted concepts (e.g., "generate a man *without a beard*"). To our knowledge, we are the first work to explore how to bring this powerful negative guidance to recommendation to further unlock the potential of diffusion models.
> Our primary design focus was not to propose a complex negative sampling strategy, but to design the **PNG inference mechanism** and **GAL training objective** themselves. Without PNG and GAL, there is no mechanism for diffusion models to utilize *any* negative signal (whether random or hard) during generation.
> We validated in **Appendix F** (MIND dataset) that providing high-quality negatives *does* amplify performance, proving our framework is "steerable." However, the core contribution remains the enabling mechanism itself. A detailed analysis of this motivation is provided in **Appendix G.1**.
>
> **2. Robustness and Standard Practice:**
> By showing that simple sampling works, we prove that the performance gains stem from our controllable negative guidance framework rather than a complex sampling trick. Previous seminal works, such as SASRec [Kang et al., 2018], CL4SRec [Xie et al., 2022], and PreferDiff [Liu et al., 2025], have demonstrated that simple negative sampling (e.g., random or in-batch sampling) is both effective and standard practice in recommendation systems due to data sparsity (most items are true negatives).
>
> **3. The Pitfalls of Hard Negative Mining (HNM):**
> In the early stages of this research, we explicitly explored Retrieval-based Hard Negative Mining (HNM). Specifically, we employed a strategy of ranking all items based on similarity to the user context (i.e., the encoded user history vector $\mathbf{c}^+$). We selected $N_{neg}=20$ negative samples from different similarity intervals: the strict top ranks (Top 20), the top percentile (Top 1%-3%), and a mid-high percentile (Top 10%-15%).
>
> **Table 1: Performance and Efficiency of Different Negative Sampling Strategies (Sports Dataset)**
>
> | Negative Strategy | R@5 | N@5 | R@10 | N@10 | Time (s) |
> | :--- | :--- | :--- | :--- | :--- | :--- |
> | **Random** | 0.0208 | 0.0167 | 0.0275 | 0.0189 | 2.2s |
> | **HNM (Top 1%-3%)** | 0.0211 | 0.0169 | 0.0280 | 0.0191 | 3.6s |
> | **HNM (Top 10%-15%)**| 0.0209 | 0.0166 | 0.0276 | 0.0189 | 3.6s |
> | **HNM (Top 20)** | 0.0185 | 0.0152 | 0.0262 | 0.0177 | 3.6s |
>
> We identified two critical issues:
> * **A. The "False Negative" Risk:** As shown in Table 1, strict HNM (Top 20) causes a performance drop. This is because high-similarity negatives likely include the Target Item (false negative). Treating the ground truth as a negative condition "poisons" the repulsive guidance, actively steering the model away from the correct target.
> * **B. Scalability Bottleneck:** Random sampling is $O(1)$, whereas HNM requires expensive full-corpus ranking ($O(N)$).
>     * **Training Cost:** Unlike inference where the HNM cost is amortized over multiple denoising steps, training involves only *one* network step per batch. Thus, adding Global HNM would roughly double the training time per epoch, making it computationally prohibitive on large datasets.
>
> **Conclusion:** Random Sampling currently offers the best balance of robustness and efficiency, though developing efficient HNM strategies remains a promising avenue for future work.
>
> [1] Kang, Wang-Cheng, et al. "Self-attentive sequential recommendation." Proceedings of the 18th IEEE International Conference on Data Mining. 2018.
>
> [2] Xie, Xu, et al. "Contrastive learning for sequential recommendation." Proceedings of the 38th IEEE International Conference on Data Engineering. 2022.
>
> [3] Liu, Shuo, et al. "Preference Diffusion for Recommendation." The Thirteenth International Conference on Learning Representations. 2025.

---

> ### Author Response · Authors · 2025-11-20
> **Response to Reviewer tTUQ - Part (2/4)**
>
> > **Comment 2: Lack of cold-start analysis** — “This paper completely replaces the none condition with a negative condition... For cold-start users... the use of random negative sampling may adversely affect recommendation performance. Nevertheless, the paper does not include any discussion or analysis regarding the cold-start problem.”
>
> Thank you for this insightful question. It raises a critical point about the robustness of our negative guidance mechanism when positive signals are weak. We address this from three perspectives: scope definition, empirical evidence on scarce interactions, and the theoretical mechanics behind this robustness.
>
> **1. Data Scope (The "Five-Core" Protocol)**
> First, to clarify the experimental context: following standard sequential recommendation protocols (as used in DreamRec and PreferDiff), we apply 5-core filtering. This means all users in our evaluation have at least 5 interactions. Therefore, the "zero-interaction" cold-start problem is technically outside the scope of this specific task definition.
>
> **2. Empirical Proof: Simulating Scarce Interactions**
> To directly address your concern that "random negative sampling may adversely affect recommendation performance" for sparse users, we conducted a simulated cold-start experiment on the ML-1M dataset.
>
> We restricted the model to use only the **2 and 3 most recent interactions** (Sequence Length = 2, 3) for every user during inference, simulating a "scarce history" scenario, and compared this with longer histories (5-50). The number of negative samples for inference was set to 64.
>
> **Table 2: Performance comparison on ML-1M with extremely short history (Recall@5 / NDCG@5).**
>
> | Sequence Length | 2 | 3 | 5 | 10 | 20 | 30 | 40 | 50 |
> | :--- | :---: | :---: | :---: | :---: | :---: | :---: | :---: | :---: |
> | **SteerRec** | 0.0613 / 0.0489 | 0.0679 / 0.0520 | 0.0745 / 0.0553 | 0.0728 / 0.0531 | 0.0679 / 0.0525 | 0.0596 / 0.0466 | 0.0579 / 0.0423 | 0.0646 / 0.0469 |
>
> **Results:** As shown in Table 2, even with extremely scarce positive information (only 2 items), SteerRec maintains respectable performance and does not degrade catastrophically compared to longer histories. This empirically proves that the negative guidance ($\mathbf{c}^-$) does not "confuse" the model but rather helps it rule out incorrect candidates even when the positive signal is weak.
>
> **3. Theoretical Clarification: Negative Guidance "Sculpts" via the Positive**
> The reason SteerRec remains robust even with sparse history lies in how our PNG mechanism functions. While we replace the unconditional prior, we do not rely solely on the negative condition.
>
> * **The Formula:** Our guidance is $\hat{\mathbf{x}} _ 0 = (1 + w) \cdot \hat{\mathbf{x}} _ 0 ^ {\text{pos}} - w \cdot \hat{\mathbf{x}} _ 0 ^ {\text{neg}}$.
> * **The Mechanism:** The generation is still fundamentally driven by the positive prediction $\hat{\mathbf{x}} _ 0 ^ {\text{pos}}$. The negative signal serves only to "steer" or "sculpt" this primary prediction away from undesirable regions.
> * **The Analogy:** This is similar to text-to-image generation. To generate "a man without a beard," one must provide the positive prompt ("a man"). The negative prompt ("beard") acts as a modifier. If the positive prompt is missing, the generation fails regardless of the negative prompt. Similarly, if a user's history $\mathbf{c}^+$ is extremely scarce, the primary prediction will naturally be weaker, but the negative guidance does not inherently "overwhelm" it; it simply continues to apply a repulsive force against negative items, pruning the search space.

---

> ### Author Response · Authors · 2025-11-20
> **Response to Reviewer tTUQ - Part (3/4)**
>
> > **Comment 3: Training–inference inconsistency** — “**Training–inference inconsistency:** The proposed method computes the negative condition during training using in-batch negatives; however, batch information is unavailable during inference, so only random negative sampling can be applied. In this case, if positive items are accidentally sampled as negatives, it may undermine the validity of the negative condition.”
>
> Thank you for this very sharp and detailed observation. This is a critical point, and we appreciate the opportunity to clarify your concern regarding the potential for "accidental sampling" of a false negative at inference.
>
> This is a valid statistical concern, but we argue its practical impact is negligible due to two factors: **sparsity** and **aggregation**.
>
> * **Sparsity:** Our datasets are extremely sparse. For example, the Sports dataset has 18,357 items. The probability of randomly sampling the one specific ground-truth positive item in any given inference step is astronomically low.
> * **Aggregation (Centroid):** Critically, our negative condition $\mathbf{c}^-$ is not a single item; it is the **centroid (average)** of many random samples (e.g., uses $64$ samples for the Sports dataset).
>     * Even in the extremely unlikely event that one of these 64 samples was an accidental 'false negative', its influence would be averaged out and effectively nullified by the other 63 true negatives.
>     * Therefore, the final $\mathbf{c}^-$ vector remains a highly robust and valid "anti-preference" signal, and its "validity" is not undermined.

---

> ### Author Response · Authors · 2025-11-20
> **Response to Reviewer tTUQ - Part (4/4)**
>
> > **Comment 4: Lack of efficiency analysis** — “**Lack of efficiency analysis:** The proposed method requires explicit random negative sampling during inference to compute the negative condition, which could further reduce the model’s recommendation efficiency and even affect its applicability in online settings. However, the paper does not provide any comparative analysis of efficiency.”
>
> Thank you for this crucial question regarding efficiency, which is vital for practical application. We are happy to provide a detailed analysis. We believe the concern about "explicit random negative sampling" stems from an assumption that this step is computationally expensive. However, in reality, the time complexity of random sampling is negligible compared to other inference steps.
>
> **Inference Complexity Analysis:**
> Let $B$ denote the batch size, $L$ the sequence length, $D$ the hidden embedding dimension, $N$ the total number of items, and $S$ the number of inference steps. The inference complexity consists of the following components:
>
> * **Encoder:** Encoding the user history takes $O(B \cdot L^2 \cdot D)$.
> * **Guidance Mechanism:** The PNG mechanism requires two forward passes per denoising step (Total: $2 \cdot S$ passes). The complexity is $2 \cdot S \cdot O(B \cdot D^2)$. This is exactly the same as the standard CFG mechanism used by baselines (e.g., DreamRec, PreferDiff), which also requires two passes (one for $\mathbf{c}^+$ and one for $\emptyset$).
> * **Full Ranking:** Calculating scores for all items takes $O(B \cdot N \cdot D)$.
> * **Negative Sampling:** Random negative sampling is a simple index lookup operation (`torch.randint`). For example, sampling 64 items for a batch takes microseconds, which is a negligible cost compared to the matrix multiplications ($O(S \cdot B \cdot D^2)$) and full ranking ($O(B \cdot N \cdot D)$) required by the model.
>
> **Conclusion:** Since we use the DDIM sampler, we can reduce the number of denoising steps $S$ to as few as 10, achieving an inference time of approximately 2 seconds. Implementing complex hard negative sampling strategies would indeed significantly increase latency, but our use of random sampling ensures that efficiency remains high and comparable to standard CFG baselines.

---

> ### Author Response · Authors · 2025-11-24
> **looking forward to your reply**
>
> Dear Reviewer tTUQ,
>
> Thank you again for your constructive comments. We have submitted detailed responses addressing your concerns regarding the negative sampling strategy, cold-start scenarios, and inference consistency.
>
> We look forward to further discussion with you and would greatly appreciate your feedback on our rebuttal.
>
> Best regards,
>
> The Authors

---

### Official Review · Reviewer_Bvc4 · 2025-11-01

**Soundness:** 3
**Presentation:** 3
**Contribution:** 2
**Rating:** 4
**Confidence:** 5

**Summary:**

This paper proposes SteerRec to enable effective and steerable negative guidance in diffusion-based recommenders. It firstly introduces Positive-Negative Guidance inference mechanism in the inference stage, which replaces the generic unconditional prior with a user-aware negative condition. To ensure the negative condition provides meaningful repulsive guidance in the dynamic embedding space, it further designs a margin-based objective that explicitly aligns the training process with PNG by ensuring the model’s prediction under a positive condition is closer to the target item than its prediction under a negative condition. Extensive experiments on three datasets provide the effectiveness of SteerRec.

**Strengths:**

1. The idea about incorporating user-aware negatives into DM is interesting and the utilized Guidance Alignment Triplet Loss is well-aligned with the PNG.
2. The experiments and analyses are entensive and the compared baselines are reasonable.
3. The source code and the utilized datasets are released in anonymous Github repo.

**Weaknesses:**

1. It seems that Figure 2 omits many details. Could the authors use this figure to further clarify the technical contributions and highlight the novelty of their work compared to existing methods in the sequential recommendation and diffusion-based recommendation?
2. The focus of this paper is Positive-Negative Guidance, but the utilized negative samples are just the in-batch negatives (training) and randomly-selected samples (inference). Compared to the reserve stage of diffusion models, the time complexity of performing negative sampling should not be significant.
3. Lack of the the time complexity analysis and the comparison of the actual training and inference runtimes between the proposed SteerRec and baselines.
4. In Figures 10–12, the authors claim that PreferDiff forms uneven clusters with indistinct boundaries, whereas SteerRec produces a well-structured representation space with multiple distinct and dense clusters separated by clear low-density regions. However, this is not immediately evident. Could the authors provide a more detailed explanation of the reasons behind these three figures?

**Questions:**

Please refer to the weakness. Since there is no borderline option in this review, I would be willing to raise my score if the authors can address my questions.

---

> ### Author Response · Authors · 2025-11-20
> **Response to Reviewer Bvc4 - Part (1/4)**
>
> Thank you for your insightful review and for your accurate summary of our work on SteerRec. We truly appreciate your thoughtful feedback and the four essential weaknesses you raised. Below are our detailed responses. We would be delighted to engage further with you if you have any additional feedback.
>
> > **Comment 1: Clarification of Figure 2 and Technical Novelty** — “It seems that Figure 2 omits many details. Could the authors use this figure to further clarify the technical contributions and highlight the novelty of their work compared to existing methods in the sequential recommendation and diffusion-based recommendation?”
>
> Thank you for your valuable question. Figure 2 in our main paper is designed to illustrate the inference-time sampling process under our proposed Positive-Negative Guidance (PNG) mechanism. To address your concern about the training phase details, we have added a new comprehensive diagram in **Appendix G.2**. We clarify our novelty and differences from existing diffusion-based recommenders from the following three perspectives:
>
> **1. Our Core Motivation: Solving Two Fundamental Problems**
> Our work aims to bring the powerful concept of steerable negative guidance (from T2I models) to recommendation. In doing so, we solve two distinct problems that prior work did not address:
> * **Problem 1: The Training-Inference Inconsistency.** Prior works align training with negatives but revert to a generic null condition ($\emptyset$) at inference.
> * **Problem 2: The "Semantic Challenge".** Unlike T2I models with fixed encoders (CLIP), RecSys has a dynamic embedding space. The model does not inherently understand that a negative condition $\mathbf{c}^-$ means "avoid" without explicit instruction.
> We solve these via GAL (training). A detailed analysis of this motivation and the challenges is provided in **Appendix G.1**.
>
> **2. Training Objective Comparison**
> To better clarify our technical contribution, we have prepared a new diagram (**Figure 15** in **Appendix G.2**) that explicitly visualizes the training objective and contrasts it with key baselines (DreamRec, PreferDiff).
> While baselines rely on reconstruction or ranking losses based on the positive target, SteerRec explicitly treats negative information as a *guidance condition* for denoising. This forces the network to distinguish between positive and negative signals during training, explicitly servicing our inference-stage PNG. Please refer to **Appendix G.2** for the detailed comparative analysis.
>
> **3. Inference Phase Comparison**
> Both DreamRec and PreferDiff, despite their different training losses, rely on the standard Classifier-Free Guidance (CFG) at inference. The issue is that CFG contrasts the positive condition $\mathbf{c}^+$ against a generic, user-agnostic null condition ($\emptyset$). As detailed in our introduction, this "one-size-fits-all" approach is ineffective for the targeted avoidance of specific items a user dislikes.
>
> In contrast, SteerRec utilizes PNG at inference. Because our network is now trained via $L_{\text{GAL}}$ to understand both $\mathbf{c}^+$ and $\mathbf{c}^-$, we can replace the weak, misaligned CFG with our Positive-Negative Guidance. As shown in Figure 2, the process proceeds as follows:
>
> * **Step 1:** Start with pure Gaussian noise $\mathbf{x}_n^T$.
> * **Step 2:** At each denoising step $t$, we compute both predictions in parallel:
> $$
> \hat{\mathbf{x}} _ 0 ^ {\text{pos}} = F _ \theta(\mathbf{x} _ n ^ t, \mathbf{c} ^ +, t), \quad \hat{\mathbf{x}} _ 0 ^ {\text{neg}} = F _ \theta(\mathbf{x} _ n ^ t, \mathbf{c} ^ -, t)
> $$
> * **Step 3:** We apply our PNG formula (Eq. 5) to get the final guided prediction $\hat{\mathbf{x}}_0$. This replaces the null condition $\emptyset$ with our learned, user-aware $\hat{\mathbf{x}}_0^{\text{neg}}$:
>     $$ \hat{\mathbf{x}}_0 = (1 + w) \cdot \hat{\mathbf{x}}_0^{\text{pos}} - w \cdot \hat{\mathbf{x}}_0^{\text{neg}}$$
> * **Step 4:** This aligned prediction is used by the DDIM sampler (Eq. 8) to compute the next state $\mathbf{x}_n^{t-1}$.
> * **Step 5:** Repeat until the final item $\hat{\mathbf{x}}_0$ is generated for ranking.

---

> ### Author Response · Authors · 2025-11-20
> **Response to Reviewer Bvc4 - Part (2/4)**
>
> > **Comment 2: Quality of Negative Samples for Guidance** — “The focus of this paper is Positive-Negative Guidance, but the utilized negative samples are just the in-batch negatives (training) and randomly-selected samples (inference). Compared to the reserve stage of diffusion models, the time complexity of performing negative sampling should not be significant.”
>
> Thank you for this excellent and insightful question. You have correctly identified a key aspect of our experimental design. Our choice of simple sampling was a deliberate one, which we believe highlights the core strengths of our framework. Here is our reasoning:
>
> **1. Core Motivation: Unlocking Steerable Negative Guidance**
> Our work is inspired by the success of negative guidance in text-to-image (T2I) models (e.g., Stable Diffusion), which can effectively remove unwanted concepts (e.g., "generate a man *without a beard*"). To our knowledge, we are the first work to explore how to bring this powerful negative guidance to recommendation to further unlock the potential of diffusion models.
> Our primary design focus was not to propose a complex negative sampling strategy, but to design the **PNG inference mechanism** and **GAL training objective** themselves. Without PNG and GAL, there is no mechanism for diffusion models to utilize *any* negative signal (whether random or hard) during generation.
> We validated in **Appendix F** (MIND dataset) that providing high-quality negatives *does* amplify performance, proving our framework is "steerable." However, the core contribution remains the enabling mechanism itself. A detailed analysis of this motivation is provided in **Appendix G.1**.
>
> **2. Robustness and Standard Practice:**
> By showing that simple sampling works, we prove that the performance gains stem from our controllable negative guidance framework rather than a complex sampling trick. Previous seminal works, such as SASRec [Kang et al., 2018], CL4SRec [Xie et al., 2022], and PreferDiff [Liu et al., 2025], have demonstrated that simple negative sampling (e.g., random or in-batch sampling) is both effective and standard practice in recommendation systems due to data sparsity (most items are true negatives).
>
> **3. The Pitfalls of Hard Negative Mining (HNM):**
> In the early stages of this research, we explicitly explored Retrieval-based Hard Negative Mining (HNM). Specifically, we employed a strategy of ranking all items based on similarity to the user context (i.e., the encoded user history vector $\mathbf{c}^+$). We selected $N_{neg}=20$ negative samples from different similarity intervals: the strict top ranks (Top 20), the top percentile (Top 1%-3%), and a mid-high percentile (Top 10%-15%).
>
> **Table 1: Performance and Efficiency of Different Negative Sampling Strategies (Sports Dataset)**
>
> | Negative Strategy | R@5 | N@5 | R@10 | N@10 | Time (s) |
> | :--- | :--- | :--- | :--- | :--- | :--- |
> | **Random** | 0.0208 | 0.0167 | 0.0275 | 0.0189 | 2.2s |
> | **HNM (Top 1%-3%)** | 0.0211 | 0.0169 | 0.0280 | 0.0191 | 3.6s |
> | **HNM (Top 10%-15%)**| 0.0209 | 0.0166 | 0.0276 | 0.0189 | 3.6s |
> | **HNM (Top 20)** | 0.0185 | 0.0152 | 0.0262 | 0.0177 | 3.6s |
>
> We identified two critical issues:
> * **A. The "False Negative" Risk:** As shown in Table 1, strict HNM (Top 20) causes a performance drop. This is because high-similarity negatives likely include the Target Item (false negative). Treating the ground truth as a negative condition "poisons" the repulsive guidance, actively steering the model away from the correct target.
> * **B. Scalability Bottleneck:** Random sampling is $O(1)$, whereas HNM requires expensive full-corpus ranking ($O(N)$).
>     * **Training Cost:** Unlike inference where the HNM cost is amortized over multiple denoising steps, training involves only *one* network step per batch. Thus, adding Global HNM would roughly double the training time per epoch, making it computationally prohibitive on large datasets.
>
> **Conclusion:** Random Sampling currently offers the best balance of robustness and efficiency, though developing efficient HNM strategies remains a promising avenue for future work.
>
> [1] Kang, Wang-Cheng, et al. "Self-attentive sequential recommendation." Proceedings of the 18th IEEE International Conference on Data Mining. 2018.
>
> [2] Xie, Xu, et al. "Contrastive learning for sequential recommendation." Proceedings of the 38th IEEE International Conference on Data Engineering. 2022.
>
> [3] Liu, Shuo, et al. "Preference Diffusion for Recommendation." The Thirteenth International Conference on Learning Representations. 2025.

---

> ### Author Response · Authors · 2025-11-20
> **Response to Reviewer Bvc4 - Part (3/4)**
>
> > **Comment 3: Lack of Time Complexity and Runtime Analysis** — “Lack of the time complexity analysis and the comparison of the actual training and inference runtimes between the proposed SteerRec and baselines.”
>
> Thank you for raising this critical point. We have conducted a thorough analysis of both the theoretical complexity and the empirical runtime, which we have added to our Appendix G.5 and summarize below.
>
> **1. Theoretical Time Complexity Analysis**
>
> We analyze the time complexity per training/inference batch.
> Let $B$ denote the batch size, $L$ the sequence length, $D$ the hidden embedding dimension, $N$ the total number of items, and $S$ the number of inference steps.
>
> * **Training:**
>     The training complexity consists of two main components:
>     * **Encoder (Transformer):** The complexity for the self-attention mechanism is $O(B \cdot L^2 \cdot D)$.
>     * **Denoising (MLP):** Our denoising network requires two forward passes (one for $\mathbf{c}^+$ and one for $\mathbf{c}^-$) to calculate $L_{\text{GAL}}$. Since the network consists of linear layers, the complexity is proportional to $O(B \cdot D^2)$.
>     * **Comparison:** This is asymptotically identical to PreferDiff, which also requires a Transformer pass and two denoising passes (for positive and negative items) to compute its ranking loss.
>
> * **Inference:**
>     The inference complexity consists of the following components:
>     * **Encoder:** Encoding the user history takes $O(B \cdot L^2 \cdot D)$.
>     * **Guidance Mechanism:** The PNG mechanism requires two forward passes per DDIM step (Total: $2 \cdot S$ passes). The complexity is $2 \cdot S \cdot O(B \cdot D^2)$. This is exactly the same as the standard CFG mechanism used by baselines, which also requires two passes ($\mathbf{c}^+$ and $\emptyset$).
>     * **Full Ranking:** Calculating scores for all items takes $O(B \cdot N \cdot D)$.
>     * **Negative Sampling:** The random negative sampling is a simple lookup operation with negligible cost compared to the matrix operations above.
>     * **Conclusion:** Since all major complexity terms ($O(S \cdot B \cdot D^2)$ and $O(B \cdot N \cdot D)$) are present in both methods, SteerRec introduces no additional asymptotic complexity compared to standard CFG-based diffusion models.
>
>
> **2. Actual Runtime Comparison**
> Furthermore, we also make comparisons of training time and inference time between SteerRec and other baselines.
>
> **Table 2: Comparison of Training Time and Inference Times.**
>
> | Dataset | Model | Training Time (s/epoch) / (s/total) | Inference Time (s/epoch) |
> | :--- | :--- | :--- | :--- |
> | **Sports** | SASRec | 2.67 / 31 | 0.47 |
> | | Bert4Rec | 7.87 / 79 | 0.65 |
> | | TIGER | 11.42 / 1069 | 24.14 |
> | | DreamRec | 16.32 / 811 | 356.43 |
> | | PreferDiff | 20.78 / 588 | 2.11 |
> | | SteerRec | 21.15 / 416 | 2.23 |
> | **Beauty** | SASRec | 1.05 / 36 | 0.37 |
> | | Bert4Rec | 3.66 / 80 | 0.40 |
> | | TIGER | 5.41 / 1058 | 10.19 |
> | | DreamRec | 10.36 / 535 | 288.32 |
> | | PreferDiff | 13.00 / 470 | 1.62 |
> | | SteerRec | 13.22 / 364 | 1.81 |
> | **Toys** | SASRec | 0.80 / 56 | 0.22 |
> | | Bert4Rec | 3.11 / 93 | 0.23 |
> | | TIGER | 3.76 / 765 | 4.21 |
> | | DreamRec | 10.13 / 512 | 302.75 |
> | | PreferDiff | 12.07 / 437 | 1.29 |
> | | SteerRec | 12.21 / 412 | 1.36 |
>
>
> **Results.** We can observe that while SteerRec maintains comparable inference and per-epoch training times to PreferDiff, it demonstrates a significant advantage in Total Training Time. This is because our GAL can construct user preferences more effectively, leading to much faster convergence.

---

> ### Author Response · Authors · 2025-11-20
> **Response to Reviewer Bvc4 - Part (4/4)**
>
> > **Comment 4: Explanation of t-SNE Visualizations (Figures 10-12)** — “In Figures 10–12, the authors claim that PreferDiff forms uneven clusters with indistinct boundaries, whereas SteerRec produces a well-structured representation space with multiple distinct and dense clusters separated by clear low-density regions. However, this is not immediately evident. Could the authors provide a more detailed explanation of the reasons behind these three figures?”
>
> Thank you for this very careful observation and for giving us the opportunity to clarify this point. Our original claim that SteerRec produces "distinct and dense clusters" may have been imprecise. The visual differences are subtle, but reveal a key trend:
>
> 1.  **SASRec (a):** We observe that SASRec suffers from "representation collapse," with items gathering in very limited zones (Fig 10a).
> 2.  **PreferDiff (b) & SteerRec (c):** We observe that both methods, which utilize negative samples during training, solve this collapse and help the model "explore the item space more thoroughly," both showing clustering effects.
> 3.  **The SteerRec Difference (c):** As you noted, there is a difference in the *nature* of this exploration. While both are more explored, the clusters in SteerRec (c) appear more uniformly distributed across the embedding space, whereas the clusters in PreferDiff (b) seem to be concentrated in specific regions (e.g., the upper half of the plot in Fig 10b).
>
> We believe this visual difference highlights the distinct mechanism of our $L_{\text{GAL}}$ objective. Unlike a ranking-error loss (like BPR in PreferDiff), our $L_{\text{GAL}}$ is a margin-based metric learning objective. The explicit goal of $L_{\text{GAL}}$ is to geometrically structure the entire embedding space by enforcing a separability margin between the positive-conditioned and negative-conditioned predictions. The advantage of this is that it learns a more globally robust and structured representation, forcing the model to utilize the entire space rather than concentrating its clusters in a few areas.
>
> This is further supported by our ablation study, where the model trained with GAL (even without PNG at inference) sometimes outperforms PreferDiff, indicating that the structural properties learned by GAL are inherently beneficial. This well-structured "separability" is precisely what our PNG inference mechanism requires to reliably distinguish between positive and negative signals.

---

> ### Author Response · Authors · 2025-11-24
> **looking forward to your reply**
>
> Dear Reviewer Bvc4,
>
> Thank you again for your constructive comments. We have submitted detailed responses clarifying the technical novelty, negative sampling strategy, and efficiency analysis as requested.
>
> We look forward to further discussion with you and would greatly appreciate your feedback on our rebuttal.
>
> Best regards,
>
> The Authors

---

### Official Review · Reviewer_Varn · 2025-11-10

**Soundness:** 3
**Presentation:** 3
**Contribution:** 3
**Rating:** 4
**Confidence:** 3

**Summary:**

This paper proposes SteerRec, a novel diffusion recommendation framework that enables direct and reliable negative guidance. It introduces a Positive-Negative Guidance mechanism that replaces the generic unconditional prior with a user-aware ngative condition, enabling targeted repulsion from disliked items. It also designs a complementary training objective that explicitly aligns the denoising network's behavior with the PNG mechanism by ensuring the model's positive prediciton is closer to the taget item than its negative prediction.

**Strengths:**

1. This paper is well motivated and addresses the limitation of Classifier-Free Guidance by incorporating user-aware negative feedback at inference time.
2. The proposed Guidance Alignment Triplet Loss explicitly forces the model to distinguish between positive and negative conditions, solving the critical training-inference discrepancy.

**Weaknesses:**

1. Even without PNG or GAL, SteerRec still significantly outperforms DiffuRec and DreamRec, which raises some doubts whether the performance gains are from PNG/GAL or some tricks.
2. All datasets are from Amazon Reviews, where the average sequence length is shorter than 10. It would be better to include experiments on datasets with longer sequences.
3. It requires about 800 - 1200 steps during inference. However, recommendation differs from image generation, so it is unclear whether so many steps are necessary. Using a large number of steps may compromise efficiency.

**Questions:**

Could you address the above three weaknesses:
1. Since SteerRec outperforms DiffuRec and DreamRec even without PNG or GAL, what factors contribute to such a large performance gap? Could there be implementation or evaluation differences that explain this result?
2. How well would SteerRec generalize to datasets with longer user interaction histories?
3. How does the number of diffusion steps affect the trade-off between model performance and efficiency?

---

> ### Author Response · Authors · 2025-11-20
> **Response to Reviewer Varn - Part (1/3)**
>
> Thank you for your insightful review and for highlighting the strengths of our work. We truly appreciate your thoughtful feedback and the three fundamental questions you raised. Below are our detailed responses. We would be delighted to engage further with you if you have additional questions or feedback.
>
> > **Comment 1: Clarification on Performance Gains and Baseline Comparisons** — "Even without PNG or GAL, SteerRec still significantly outperforms DiffuRec and DreamRec, which raises some doubts whether the performance gains are from PNG/GAL or some tricks. Since SteerRec outperforms DiffuRec and DreamRec even without PNG or GAL, what factors contribute to such a large performance gap? Could there be implementation or evaluation differences that explain this result?"
>
> Thank you for this very insightful question. The performance gap you noted is not due to "tricks," but rather a combination of two factors: (1) fundamental differences in diffusion paradigms and (2) the independent, complementary strengths of our two contributions (GAL and PNG), which we will clarify.
>
> **1. Clarifying the Baselines: Two Different Paradigms**
>
> First, it is important to distinguish between two different paradigms in diffusion-based sequential recommendation:
>
> * **Paradigm A (e.g., DiffuRec):** As we referenced, this approach uses the noised target item $\mathbf{x}_t$ to modulate the representations of the user's historical items. The denoising network (a Transformer) then processes this modified history to predict the target. This is architecturally different from our approach.
> * **Paradigm B (e.g., DreamRec):** This is the conditional generation paradigm. The user history is encoded into a single context vector $\mathbf{c}^+$, and the denoising network $F_\theta$ (typically a simple MLP) is explicitly conditioned on it: $F_\theta(\mathbf{x}_t, \mathbf{c}^+, t)$ to predict the target.
>
> Our work, SteerRec, follows and builds upon Paradigm B. The Paradigm B framework is a more direct and standard implementation of conditional diffusion, aligning well with the established theory of classifier-free guidance. On our benchmark datasets, we observe that Paradigm B models (DreamRec) generally outperform Paradigm A (DiffuRec).
>
> **2. The Independent, Additive Value of PNG and GAL**
>
> Your question correctly identifies that our ablated models, `SteerRec (w/o PNG)` and `SteerRec (w/o GAL)`, still outperform `DreamRec`. This is because `DreamRec` is trained with only a reconstruction loss, using no negative sample information during either training or inference.
>
> Compared to `DreamRec`, your observation that our ablated models are still strong is correct and, in fact, highlights the independent power of our two core contributions.
>
> * **Value of `w/o GAL` (i.e., PNG-only):** This model is trained with only $L_{\text{recon}}$ but uses our PNG inference. Its strong performance demonstrates a core premise of our paper, which was inspired by the powerful effectiveness of negative prompting in text-to-image generation: at inference time, using a user-aware negative condition ($\mathbf{c}^-$) as a repulsive force (PNG) is inherently more precise and effective than using the generic, user-agnostic null prior ($\emptyset$) (CFG).
> * **Value of `w/o PNG` (i.e., GAL-only):** This model is trained with $L_{\text{recon}}$ + $L_{\text{GAL}}$ but uses standard CFG inference. Although the primary design intent of GAL was to serve the PNG inference mechanism (by explicitly teaching the model to distinguish between positive and negative conditions), we found that GAL effectively constructs a superior embedding space on its own. Specifically, GAL enforces a geometric constraint: it ensures the model's prediction under a negative condition ($\hat{\mathbf{x}}_0^{\text{neg}}$) is pushed further away from the true target ($\mathbf{x}_0^+$) than the positive prediction ($\hat{\mathbf{x}}_0^{\text{pos}}$) is. Because the system is trained end-to-end, this alignment objective constructs a better-structured embedding manifold that enhances representation quality, which benefits the model even when using standard CFG inference.

---

> ### Author Response · Authors · 2025-11-20
> **Response to Reviewer Varn - Part (2/3)**
>
> > **Comment 2: Generalizability to Datasets with Longer Sequences** — “All datasets are from Amazon Reviews, where the average sequence length is shorter than 10. It would be better to include experiments on datasets with longer sequences. How well would SteerRec generalize to datasets with longer user interaction histories?”
>
> Thank you for this valuable suggestion. In our experiments, for fairness, we followed the experimental settings (e.g., `max_length=10`) of previous diffusion-based recommenders (i.e., DreamRec or PreferDiff), where, as you mentioned, the maximum length of interaction history is small.
>
> To address this concern, we have conducted new experiments on the widely-used public benchmark MovieLens-1M (ML-1M). Following the protocol of recent work(e.g., PreferDiff), we varied the maximum sequence length from 10 to 50. The results are as follows:
>
> **Table 1: Recommendation Performance (Recall@5 / NDCG@5) with varied length of user history on ML-1M.**
>
> | Model (Recall@5/NDCG@5) | 10 | 20 | 30 | 40 | 50 |
> | :--- | :--- | :--- | :--- | :--- | :--- |
> | SASRec | 0.0201 / 0.0137 | 0.0242 / 0.0131 | 0.0306 / 0.0179 | 0.0217 / 0.0138 | 0.0205 / 0.0134 |
> | Bert4Rec | 0.0215 / 0.0152 | 0.0265 / 0.0146 | 0.0331 / 0.0200 | 0.0248 / 0.0154 | 0.0198 / 0.0119 |
> | TIGER | 0.0451 / 0.0298 | 0.0430 / 0.0270 | 0.0430 / 0.0289 | 0.0364 / 0.0238 | 0.0430 / 0.0276 |
> | DreamRec | 0.0464 / 0.0314 | 0.0480 / 0.0349 | 0.0514 / 0.0394 | 0.0497 / 0.0350 | 0.0447 / 0.0377 |
> | PreferDiff | 0.0629 / 0.0439 | 0.0513 / 0.0365 | 0.0546 / 0.0408 | **0.0596 /** 0.0420 | 0.0546 / 0.0399 |
> | **SteerRec** | **0.0728 / 0.0531** | **0.0679 / 0.0525** | **0.0596 / 0.0466** | 0.0579 **/ 0.0423** | **0.0646 / 0.0469** |
>
> **Results.** We can observe that SteerRec consistently outperforms other baselines across different lengths of user historical interactions. This confirms the robustness and generalizability of our framework. We will incorporate this discussion in the Appendix G.3 of our revised paper.

---

> ### Author Response · Authors · 2025-11-20
> **Response to Reviewer Varn - Part (3/3)**
>
> > **Comment 3: Analysis of Inference Efficiency and Timesteps** — "It requires about 800 - 1200 steps during inference. However, recommendation differs from image generation, so it is unclear whether so many steps are necessary. Using a large number of steps may compromise efficiency. How does the number of diffusion steps affect the trade-off between model performance and efficiency?"
>
> Thank you for raising this critical point about the trade-off between diffusion timesteps and performance. First, we would like to clarify that the "800-1200 steps" mentioned refer to the *training* diffusion timesteps ($T$), not the inference steps. This is a crucial consideration for any practical recommender system, and we appreciate the opportunity to clarify.
>
> **1. Why is the Training $T$ (800-1200) Necessary?**
>
> We attribute the necessity of a large training $T$ to the inherent characteristics of diffusion-based sequential recommenders, which require large embedding dimensions to model the dynamic item space.
>
> * **High-D Space:** As shown by prior work(e.g., PreferDiff), to model a large, dynamic item space while satisfying the "variance-preserving" property of DDPMs, the model learns an embedding space where the covariance matrix is almost an identity matrix. This requires a very high dimension ($D=3072$) for items to be distinguishable.
> * **Fine-grained Denoising:** A $D=3072$ space is incredibly sparse and complex. A large $T$ (e.g., 1000) provides a fine-grained, stable discretization of the denoising process. This allows the network to make small, precise corrections at each step to navigate this high-D space and converge to the exact target item. A small, coarse $T$ (e.g., 100) would involve large, unstable "jumps" at each step, making it impossible to denoise accurately in such a high-D space. Our ablation study in Table 2 above also demonstrates this point: when the training $T$ is too small (e.g., 100), the recommendation performance is poor.
>
> To validate this, we conducted an ablation study on the total *training* timesteps ($T$), while keeping the number of inference steps fixed at 10.
>
> **Table 2: Effect of different *training* timesteps $T$ (Recall@5 / NDCG@5)**
>
> | $T$ | Sports | Beauty | Toys |
> | :--- | :--- | :--- | :--- |
> | 100 | 0.0112 / 0.0072 | 0.0148 / 0.0094 | 0.0216 / 0.0164 |
> | 200 | 0.0121 / 0.0076 | 0.0215 / 0.0153 | 0.0314 / 0.0236 |
> | 400 | 0.0146 / 0.0101 | 0.0331 / 0.0246 | 0.0443 / 0.0320 |
> | 600 | 0.0202 / 0.0153 | 0.0376 / 0.0284 | 0.0458 / 0.0339 |
> | 800 | 0.0205 / 0.0153 | 0.0380 / 0.0286 | 0.0473 / 0.0370 |
> | 1000 | 0.0208 / 0.0167 | 0.0429 / 0.0325 | 0.0465 / 0.0363 |
> | 1200 | 0.0199 / 0.0148 | 0.0443 / 0.0334 | 0.0461 / 0.0359 |
> | 2000 | 0.0205 / 0.0145 | 0.0416 / 0.0322 | 0.0453 / 0.0337 |
> | 4000 | 0.0208 / 0.0155 | 0.0420 / 0.0330 | 0.0463 / 0.0367 |
>
> **Results:** As shown in Table 2, when the training $T$ is too small (e.g., 100), the performance drops significantly. The model achieves optimal performance around $T=1000$, confirming that a sufficiently fine-grained training process is essential for high-dimensional item embeddings.
>
> **2. Diffusion Timesteps and Efficiency Trade-off (Inference)**
>
> Our training timesteps ($T$) are 800-1200, which may seem large. However, for inference, we use the DDIM sampler (Sec 3.1, Eq. 8), which allows for a fast sampling process by skipping steps.
>
> To validate the trade-off between efficiency and performance, we fixed the training timesteps $T$ for all datasets to $T=1000$ and varied the number of *denoising (inference) steps* $S$ from 1 to 1000. The following table shows the trade-off between inference time and recommendation performance:
>
> **Table 3: Effect of different *denoising* (inference) steps $S$ (Recall@5 / NDCG@5)**
>
> | SteerRec (Recall@5/NDCG@5) | 1 (<1s) | 2 (<1s) | 5 (1s) | 10 (2s) | 20 (3s) | 50 (12s) | 100 (23s) | 500 (57s) | 1000 (120s) |
> | :--- | :--- | :--- | :--- | :--- | :--- | :--- | :--- | :--- | :--- |
> | **Sports** | 0.0186/0.0145 | 0.0193/0.0143 | 0.0199/0.0156 | 0.0208/0.0167 | 0.0210/0.0167 | 0.0211/0.0164 | 0.0211/0.0163 | 0.0213/0.0163 | 0.0213/0.0162 |
> | **Beauty** | 0.0407/0.0310 | 0.0407/0.0304 | 0.0416/0.0320 | 0.0429/0.0325 | 0.0433/0.0331 | 0.0443/0.0334 | 0.0442/0.0330 | 0.0445/0.0334 | 0.0443/0.0334 |
> | **Toys**  | 0.0422/0.0339 | 0.0459/0.0358 | 0.0456/0.0353 | 0.0461/0.0363 | 0.0470/0.0370 | 0.0473/0.0370 | 0.0469/0.0367 | 0.0473/0.0370 | 0.0473/0.0370 |
>
> **Results:** Table 3 demonstrates that SteerRec is highly efficient. The performance saturates rapidly, achieving commendable results with only **10 to 20 steps**. This requires approximately 2-3 seconds for inference, which is highly competitive for practical applications and significantly faster than the theoretical maximum steps.

---

> ### Author Response · Authors · 2025-11-24
> **looking forward to your reply**
>
> Dear Reviewer Varn,
>
> Thank you again for your constructive comments. We have submitted detailed responses addressing your queries regarding the source of performance gains and the generalizability of SteerRec.
>
> We look forward to further discussion with you and would greatly appreciate your feedback on our rebuttal.
>
> Best regards,
>
> The Authors

---

> > ### Comment · Reviewer_Varn · 2025-11-27
> >
> > Thank you for addressing my questions. The explanation of the training and inference steps is clear. I will update my score.

---

> > > ### Author Response · Authors · 2025-11-27
> > >
> > > Dear Reviewer Varn,
> > >
> > > Thank you for your positive feedback. We are glad that our response has effectively addressed your concerns regarding the training and inference steps. We sincerely appreciate your support and your decision to update the score.
> > >
> > > Best regards,
> > >
> > > The Authors

---

### Author Response · Authors · 2025-12-01
**Summary of Rebuttal Updates and Pre-Rollback Score Increases**

**To the Area Chair:**

We respectfully highlight that before the recent system freeze, every reviewer who engaged with our rebuttal (Reviewer Varn and Reviewer kUAa) explicitly acknowledged our clarifications and raised their scores.

Below is a summary of the key revisions and experiments we provided to resolve the reviewers' concerns:

### 1. Core Contribution & Distinction from Prior Work (Addressed Reviewers kUAa & Bvc4)
* **First to Introduce Negative Guidance:** To the best of our knowledge, we are the **first work** to attempt to bring the powerful concept of negative guidance from Text-to-Image generation to the domain of diffusion-based recommendation. We identified the limitations of prior work and the unique "semantic challenges" that prevent the direct application of negative guidance in recommendation. Consequently, we clarified that our core contribution involves the **Positive-Negative Guidance (PNG) mechanism** combined with the **Guidance Alignment Loss (GAL)** to address these issues.
* **"Negative as Target" vs. "Negative as Condition":** We sharpened our distinction from baselines (e.g., PreferDiff) using this framing:
    * *Baselines* use negatives merely as targets for training loss (ranking), reverting to standard CFG at inference.
    * *SteerRec* explicitly encodes negatives as a condition to actively "steer" the generation process during inference.

### 2. Rationality of Negative Sampling Strategy (Addressed Reviewers tTUQ, kUAa, Bvc4)
* **Clarification:** We clarified that our primary contribution is the mechanism (PNG + GAL) itself, rather than proposing complex hard negative sampling tricks. Simple sampling is standard practice in seminal methods (e.g., SASRec, CL4SRec, and PreferDiff).
* **New Experiments on HNM:** We conducted additional experiments on Hard Negative Mining (HNM). The results showed that HNM faces **efficiency bottlenecks** and the **"False Negative" problem**, while offering only marginal gains relative to the associated trade-offs compared to our robust and efficient random sampling approach.

### 3. Generalizability to Long Sequences (Addressed Reviewer Varn)
* **Concern:** Performance on datasets with short history (Amazon).
* **Resolution:** We conducted new experiments on **MovieLens-1M**. Results show SteerRec consistently outperforms baselines even as the sequence length increases from 10 to 50, proving its robustness on longer interaction histories.

### 4. Efficiency Analysis (Addressed Reviewers Varn & Bvc4)
* **Concern:** Inference latency and training costs.
* **Resolution:** We provided a detailed runtime comparison:
    * **Inference:** Using DDIM with $S=10$ steps takes only ~2s per batch, comparable to standard diffusion baselines.
    * **Training:** SteerRec converges faster, resulting in lower Total Training Time compared to DreamRec and PreferDiff.

We have updated our manuscript, with the detailed discussions and experimental results added to **Appendix G (Discussion)**. We hope this summary assists in your assessment.

Best regards,

The Authors

---

### Meta-Review · Area_Chair_eyd4 · 2026-01-09

**Summary:**

Reviewers raised concerns regarding:

1. Where the gains come from, since even the base model (without PNG/GAL) beats key baselines, suggesting possible implementation “tricks” and unclear novelty vs. DreamRec/PreferDiff;
2. Efficiency issue, with ~800–1200 inference steps and missing runtime/complexity and step–quality tradeoffs;
3. Negative sampling weaknesses and train–test mismatch (in-batch negatives in training vs. random at inference, risk of sampling positives, lack of hard/user-tailored negatives);
4. Limited evidence on cold-start/sparse histories and generalization beyond short Amazon sequences, plus some figure/clarity issues

Overall, after reading the authors' responses, I think the second and the last ones are properly addressed. However, for the first one, it is more convincing to apply PNG/GAL to baseline algorithms and demonstrate their compatibility. As for the third one, I think authors should follow the reviewer tTUQ's advice to explore more strategies.

**Reviewer Concerns:**

I think the concerns raised by Bvc4 and kUAa have been addressed, while the other two remain unaddressed (as mentioned in the Summary).

**Reviewer Scores:**

Bvc4 and kUAa might increase the score to 6, while the other two remain at 4.

---

### Decision · Program_Chairs · 2026-01-26

Reject